# A cellular and molecular atlas reveals the basis of chytrid development

**Davis Laundon[1,2], Nathan Chrismas[1], Kimberley Bird[1], Seth Thomas[1], Thomas Mock[2], Michael Cunliffe[1,3]***

[1]Marine Biological Association, The Laboratory, Citadel Hill, Plymouth, United Kingdom; [2]School of Environmental Sciences, University of East Anglia, Norwich, United Kingdom; [3]School of Biological and Marine Sciences, University of Plymouth, Plymouth, United Kingdom

**ABSTRACT** The chytrids (phylum Chytridiomycota) are a major fungal lineage of ecological and evolutionary importance. Despite their importance, many fundamental aspects of chytrid developmental and cell biology remain poorly understood. To address these knowledge gaps, we combined quantitative volume electron microscopy and comparative transcriptome profiling to create an 'atlas' of the cellular and molecular basis of the chytrid life cycle, using the model chytrid *Rhizoclosmatium globosum*. From our developmental atlas, we describe the transition from the transcriptionally inactive free-swimming zoospore to the more biologically complex germling, and show that lipid processing is multifaceted and dynamic throughout the life cycle. We demonstrate that the chytrid apophysis is a compartmentalised site of high intracellular trafficking, linking the feeding/attaching rhizoids to the reproductive zoosporangium, and constituting division of labour in the chytrid cell plan. We provide evidence that during zoosporogenesis, zoospores display amoeboid morphologies and exhibit endocytotic cargo transport from the interstitial maternal cytoplasm. Taken together, our results reveal insights into chytrid developmental biology and provide a basis for future investigations into non-dikaryan fungal cell biology.

*For correspondence:
micnli@mba.ac.uk

**Competing interest:** The authors declare that no competing interests exist.

## Editor's evaluation

This data rich manuscript provides highly valuable insights into chytrid development and will accelerate further research on non-dikaryan fungal cell biology.

## Introduction

The chytrids (phylum Chytridiomycota) are a major, predominantly unicellular fungal lineage of ecological importance. For example, some chytrid species are the causative agents of the global amphibian panzootic (*Fisher and Garner, 2020*) and virulent crop pests (*van de Vossenberg et al., 2019*), whilst others are algal parasites and saprotrophs in marine and freshwater ecosystems (*Frenken et al., 2017*; *Grossart et al., 2019*; *Klawonn et al., 2021*). Chytrid zoospores contain large amounts of intracellular storage lipids that are consumed by grazing zooplankton, making them responsible for a significant form of trophic upgrading in aquatic ecosystems (*Kagami et al., 2017*; *Kagami et al., 2014*; *Kagami et al., 2007*; *Rasconi et al., 2020*). Chytrids are also important from an evolutionary perspective as they retain cellular traits from the last common ancestor of branching fungi (*Figure 1A*) that are now absent in hyphal fungi (*Berbee et al., 2017*; *Nagy et al., 2018*), as well as traits from the common ancestor of animals and fungi in the Opisthokonta (*Medina et al., 2016*; *Prostak et al., 2021*). This makes chytrids powerful models to explore the origin and evolution of innovations in fungal cell biology and the wider eukaryotic tree of life. To help fully appreciate chytrids in terms of

**Figure 1.** Chytrids are a major fungal phylum with a dimorphic life cycle. (**A**) Chytrids (phylum Chytridiomycota) are a major fungal lineage, many members of which exhibit cellular characteristics retained from the last common ancestor of branching (rhizoidal and hyphal) fungi (star). Simplified phylogenetic tree from *Laundon and Cunliffe, 2021*; *Tedersoo et al., 2018*. (**B**) *Rhizoclosmatium globosum* exhibits an archetypal chytrid life cycle and cell plan delineated here into four discrete major stages. Labelled is the apophysis (**a**), cell body (**b**), and rhizoids (**r**). Scale bar = 10 μm. (**C**) Diagrammatic workflow of the experimental setup used in this study for comparative cellular serial block face scanning electron microscopy (SBF-SEM) and molecular (transcriptome) analysis.

The online version of this article includes the following figure supplement(s) for figure 1:

**Figure supplement 1.** Representative images from confocal surveys conducted to assess the synchronicity of cell cultures for serial block face scanning electron microscopy (SBF-SEM) and RNA-Seq harvesting.

their ecological and evolutionary contexts, it is necessary to resolve their core cell biology (*Laundon and Cunliffe, 2021*).

Central to chytrid cell biology is their distinctive dimorphic life cycle, consisting of a motile free-swimming uniflagellate zoospore that transforms into a sessile walled thallus with anucleate attaching and feeding rhizoids (*Figure 1B*; *Laundon et al., 2020*; *Medina et al., 2020*). The cell body component of the thallus develops into the zoosporangium from which the next generation of zoospores are produced (*Figure 1B*). Any biological life cycle inherently represents a temporal progression, yet the chytrid life cycle can be categorised into four distinctive contiguous life stages (*Berger et al., 2005*). The first stage is the motile 'zoospore' which lacks a cell wall, does not feed, and colonises substrates or hosts. Even though zoospores are metabolically active, they are transcriptionally and translationally inactive, with dormant ribosomes containing maternally derived mRNA (*Medina and Buchler, 2020*). The second stage is the sessile 'germling' which develops immediately after zoospore settlement following flagellar retraction (or sometimes detachment), cell wall production (encystment), and initiation of rhizoid growth from an initial germ tube. The third stage is the vegetative 'immature thallus' which is associated with the highest levels of rhizoid development and overall cellular growth (*Laundon et al., 2020*). The cell plan of the immature thallus can be divided into three parts (*Figure 1B*): the cell body which is ultimately destined for reproduction (zoosporogenesis), the rhizoid for attachment and feeding, and (in some chytrid species) a bulbous swelling between the cell body and rhizoid termed the 'apophysis', the function of which is currently poorly understood (*Laundon and Cunliffe, 2021*). The final life stage is the reproductive 'mature zoosporangium', which appears once the immature thallus has reached maximum cell size and the cell body cytoplasm is cleaved into the next generation of zoospores (*Figure 1B*).

Representative model strains have an important role to play in understanding the biology of chytrids (*Laundon and Cunliffe, 2021*). *Rhizoclosmatium globosum* is a widespread aquatic saprotroph and the strain *R. globosum* JEL800 has emerged as a promising model organism in laboratory investigations (*Laundon et al., 2020*; *Roberts et al., 2020*; *Venard et al., 2020*) due to an available genome (*Mondo et al., 2017*), easy axenic culture, and amenability to live-cell fluorescent microscopy (*Laundon et al., 2020*). *R. globosum* JEL800 exhibits an archetypal chytrid cell plan (*Figure 1B*) and rapid life cycle, making it a powerful model system to interrogate chytrid cellular development (*Laundon et al., 2020*; *Laundon and Cunliffe, 2021*).

The chytrid life cycle has so far been characterised with largely descriptive approaches (e.g. *Berger et al., 2005*) with only a few quantitative studies focusing on specific processes, such as rhizoid morphogenesis (*Dee et al., 2019*; *Laundon et al., 2020*) and actin formation (*Medina et al., 2020*; *Prostak et al., 2021*). These important studies provide a foundation on which to develop a quantitative approach to understand the biology of the chytrid cell plan and the drivers of the transitions between the life stages. To investigate the cellular and molecular underpinnings of the chytrid life cycle and associated cell biology, we studied the four major life stages of *R. globosum* by combining quantitative volume electron microscopy and transcriptomics (*Figure 1C*), with the addition of supplementary targeted live-cell fluorescent microscopy and lipid analysis. The aim of our approach was to quantify the cellular traits that define the major chytrid life stages and identify the biological processes that take place during the developmental transitions between them. As such, we have created a developmental 'atlas' with *R. globosum* for the archetypal chytrid life cycle, which in turn generated specific avenues for targeted investigation of important biological processes, namely lipid biology, apophysis function, and zoosporogenesis.

By culturing *R. globosum* and sampling the populations at different stages through their temporal development (0 hr zoospore, 1.5 hr germling, 10 hr immature thallus, and a 24 hr mixed population including mature zoosporangia) (*Figure 1C*; *Figure 1—figure supplement 1*), we examined chytrid populations with both 3D reconstructions by serial block face scanning electron microscopy (SBF-SEM) and mRNA sequencing. We used single-cell SBF-SEM reconstructions (n = 5 cells per life stage) to quantify the cellular structures at each life stage and population-level transcriptomic analysis of significant Kyoto Encyclopedia of Genes and Genomes (KEGG) pathway categories (n = 3 populations per life stage, differentially expressed genes [DEGs]) to summarise the major biological differences between the life stages through temporal development (*Figure 1C*). As these stages represent key time points in the progression of the linear temporal chytrid life cycle, pairwise comparison of transcriptomes from contiguous life stages achieved an account of the major putative biological transitions

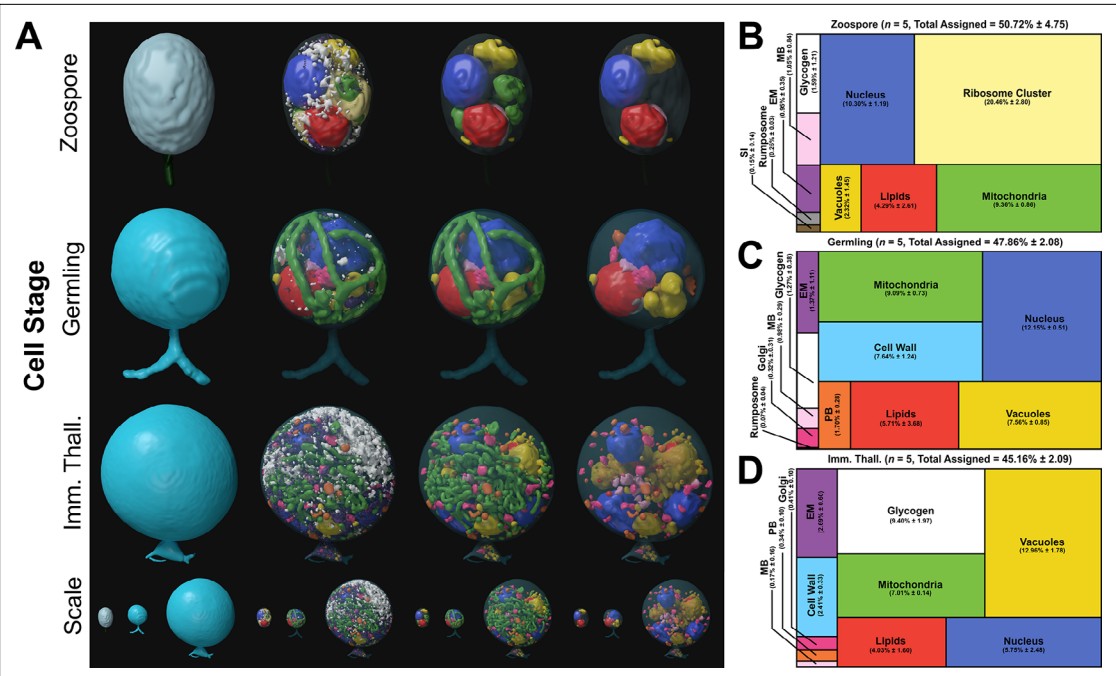

**Figure 2.** Serial block face scanning electron microscopy (SBF-SEM) reconstructions of the *Rhizoclosmatium globosum* life cycle. (**A**) Representative SBF-SEM reconstructions of the first three life stages of the *R. globosum* life cycle. Bottom row shows the stages to scale. Organelle colours as in (**B–D**) and conserved throughout. (**B–D**) Volumetric composition of assigned organelles in SBF-SEM reconstructions (n = 5) of zoospores (**B**), germlings (**C**), and immature thalli (**D**). EM = endomembrane, MB = microbodies, PB = peripheral bodies, SI = striated inclusion.

The online version of this article includes the following source data and figure supplement(s) for figure 2:

**Source data 1.** Data associated with *Figure 2*.

**Figure supplement 1.** Workflow of the image analysis protocol used to generate and visualise 3D reconstructions of chytrid cells from serial block face scanning electron microscopy (SBF-SEM) stacks.

**Figure supplement 2.** Examples of subcellular components identified in this study taken from single serial block face scanning electron microscopy (SBF-SEM) slices.

**Figure supplement 3.** Individual 3D serial block face scanning electron microscopy (SBF-SEM) reconstructions of *Rhizoclosmatium globosum* cells (*not to scale*) across life stages labelled with replicate IDs.

**Figure supplement 4.** Individual volumetric compositions of assigned organelles from *Rhizoclosmatium globosum* serial block face scanning electron microscopy (SBF-SEM) reconstructions across life stages labelled with replicate IDs.

**Figure supplement 5.** Comparisons of volumetric proportions of subcellular structures across chytrid life stages (n = 5).

(e.g. germling vs. zoospore, immature thallus vs. germling). Our findings provide insights into chytrid developmental processes and serve as a resource from which to resolve the biology of this ecologically and evolutionary important fungal lineage.

## Results

### A cellular and molecular atlas of *R. globosum* development

The orientation, subcellular localisation, and morphology of the cellular ultrastructure determined with SBF-SEM of the *R. globosum* zoospore, germling, and immature thallus life stages are shown in *Figure 2A*; *Video 1*; *Figure 2—figure supplements 1–5*, with the volumetric transition from zoospore (20.7 ± 1.7 µm³, mean ± SD) to germling (33.0 ± 2.0 µm³) to immature thallus (1116.3 ± 206.2 µm³) exceeding an order of magnitude. These ultrastructural differences in the cell patterns at each life stage (*Figure 2B–D*) are complemented with differential gene expression analysis focusing on characterising the transitions between life stages (*Figure 3*; *Figure 3—figure supplements 1–5*). Full statistical details of cell volumetric and molecular comparisons are provided in supplementary information. As the mature zoosporangium samples were taken from a mixed population of cell stages

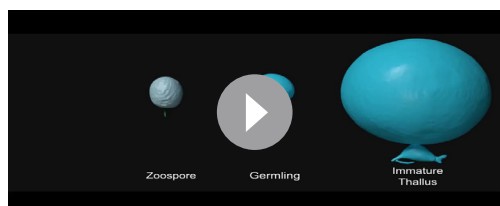

**Video 1.** Serial block face scanning electron microscopy (SBF-SEM) reconstructions allowed the structural comparison of life stages in *Rhizoclosmatium globosum*. Representative SBF-SEM reconstructions of the zoospore, germling, and immature thallus life stages for comparison. Zoospore and germling cells shown to scale at the beginning of the video, and later enlarged.

https://elifesciences.org/articles/73933/figures#video1

(*Figure 1C*), they were conservatively excluded from comparison with the first three stages and are treated separately in this analysis.

The zoospore cell body is a prolate spheroid with a posterior flagellum and is volumetrically dominated by a structurally distinct ribosome cluster (20.5% ± 2.8 %) in the cell interior which was not detected in the other life stages (*Figure 2A–B*). There was no observed significant molecular signature associated with elevated protein synthesis in zoospores, and relative to the mRNA present in the zoospore life stage, the germling transcriptome showed a downregulation of ribosome and ribosome biogenesis KEGG categories (*Figure 3C*). Only two other KEGG categories were downregulated in germlings relative to the mRNA present in zoospores. These were linked to peroxisomes and ATP-binding cassette (ABC) transporters (*Figure 3C*), both of which are associated with lipid metabolism (discussed further below).

Following encystment, the germling stage marks the origin of a complete cell wall, Golgi apparatuses (three out of five replicates), and peripheral bodies, that is, vesicular structures bound to the cell periphery putatively associated with cell wall deposition (*Figure 2A, C*; *Figure 2—figure supplement 2*), as well as the beginning of rhizoid growth from a posterior germ tube. Transcriptome analysis indicated that the germling exhibits a greater range of active processes compared to the zoospore, with upregulation of primary and secondary metabolism (e.g. amino acid and secondary metabolite biosynthesis), feeding and energy release (e.g. carbon metabolism and tricarboxylic acid cycle), and the activation of transcription and translation (e.g. spliceosome and aminoacyl-tRNA biosynthesis) KEGG categories (*Figure 3D*). A similar pattern is shown when comparing KEGG categories downregulated in immature thalli relative to germlings (*Figure 3E*). In the germling stage, we also show upregulation of genes associated with proteasome activity (*Figure 3D*). Taken together, these data characterise the transition from zoospore to germling with the activation of diverse biological processes including central metabolic pathways, cellular anabolism, and feeding.

Compared to the germling, immature thalli devoted a smaller volumetric proportion to the cell wall (IT 2.4% ± 0.3% vs. G 7.6% ± 1.2%, p < 0.01) and peripheral bodies (IT 0.3% ± 0.1% vs. G 1.7% ± 0.3%, p < 0.01) (*Figure 2A, D*; *Figure 2—figure supplement 5*). Similarly, nuclei (IT 4.8% ± 2.5% vs. G 12.2% ± 0.5%, p < 0.01) and mitochondria (IT 7.0% ± 0.1% vs. G 9.1% ± 0.7%, p < 0.001) occupied a smaller volumetric proportion (*Figure 2A, D*; *Figure 2—figure supplement 5*). Conversely, immature thalli displayed larger glycogen stores (IT 9.4% ± 2.0% vs. G 1.3% ± 0.4%, p < 0.01) and vacuole fractions (IT 13.0% ± 1.8% vs. G 7.6% ± 0.9%, p < 0.001) than germlings (*Figure 2A, D*; *Figure 2—figure supplement 5*).

The immature thalli transcriptomes showed the upregulation of KEGG categories including endocytosis and phagosomes relative to germlings (*Figure 3F*). Within these categories were genes related to microtubules and actin, including actin-related proteins-2/3 (ARP2/3), suggesting that the immature thalli are associated with higher cytoskeletal activity compared to germlings. Some immature thallus replicates were multinucleate (1.8 ± 1.3 nuclei per cell), indicating the onset of nuclear division (*Figure 2A*), which is supported by the upregulation of cell cycle and DNA replication KEGG categories relative to germlings. The apophysis (12.2 ± 6.0 μm³) was observed at the immature thallus stage (discussed further below). Overall, these data characterise the biological shift from germling to immature thallus, a transition from initiating general metabolic activity to intracellular trafficking and the start of zoosporogenesis.

As anticipated, the SBF-SEM reconstructions showed that the zoospore is wall-less unlike the germling and immature thallus stages (*Figure 2*). Single-cell fluorescent labelling of chitin (the primary wall component) however showed that precursory material is produced by zoospores at the posterior pole near the flagellum base (*Figure 4A*) suggesting that cell wall production is initiated to some extent

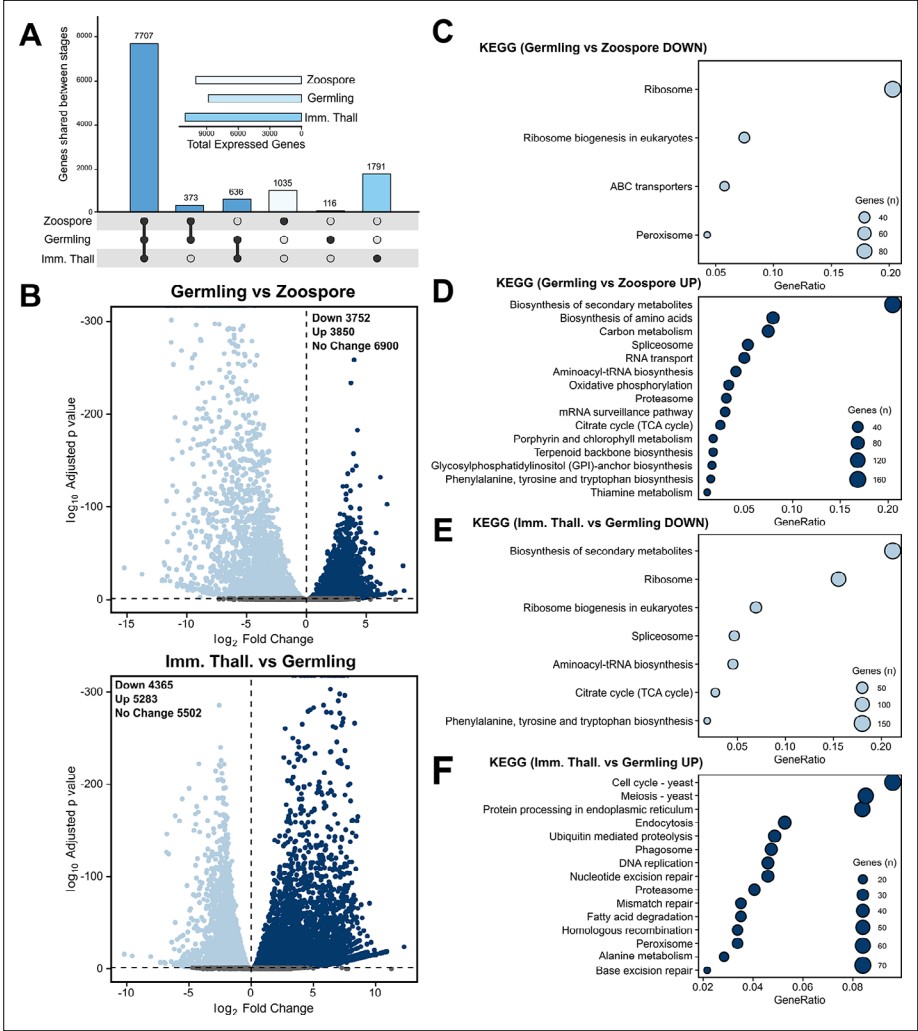

**Figure 3.** Transcriptome analysis of the *Rhizoclosmatium globosum* life cycle. (**A**) Shared and unique gene mRNA counts between life stages. Inset shows total gene mRNA counts per life stage. (**B**) Pairwise comparison of differentially expressed genes (DEGs) between germlings and zoospores, and immature thalli and germlings. (**C–F**) Pairwise comparison of significant (n = 3, p < 0.05) Kyoto Encyclopedia of Genes and Genomes (KEGG) categories between germlings and zoospores (**C–D**), and immature thalli and germlings (**E–F**).

The online version of this article includes the following source data and figure supplement(s) for figure 3:

**Source data 1.** Data associated with *Figure 3*.

**Figure supplement 1.** Heatmap clustering of all differentially expressed genes (DEGs) between zoospore, germling, and immature thallus replicates.

**Figure supplement 2.** Gene Ontology (GO) categories significantly downregulated (p < 0.05) in germlings relative to zoospores.

**Figure supplement 3.** Gene Ontology (GO) categories significantly upregulated (p < 0.05) in germlings relative to zoospores.

**Figure supplement 4.** Gene Ontology (GO) categories significantly downregulated (p < 0.05) in immature thalli relative to germlings.

**Figure supplement 5.** Gene Ontology (GO) categories significantly upregulated (p < 0.05) in immature thalli relative to germlings.

during the free-swimming zoospore stage of the *R. globosum* life cycle. In a previous study (**Laundon et al., 2020**), we identified 28 candidate genes for glycosyltransferase (GT2) domain-containing proteins putatively involved in chitin synthesis in *R. globosum* and searched for their mRNA sequences in the transcriptome data. There was no clear pattern of differential abundance of mRNA sequences

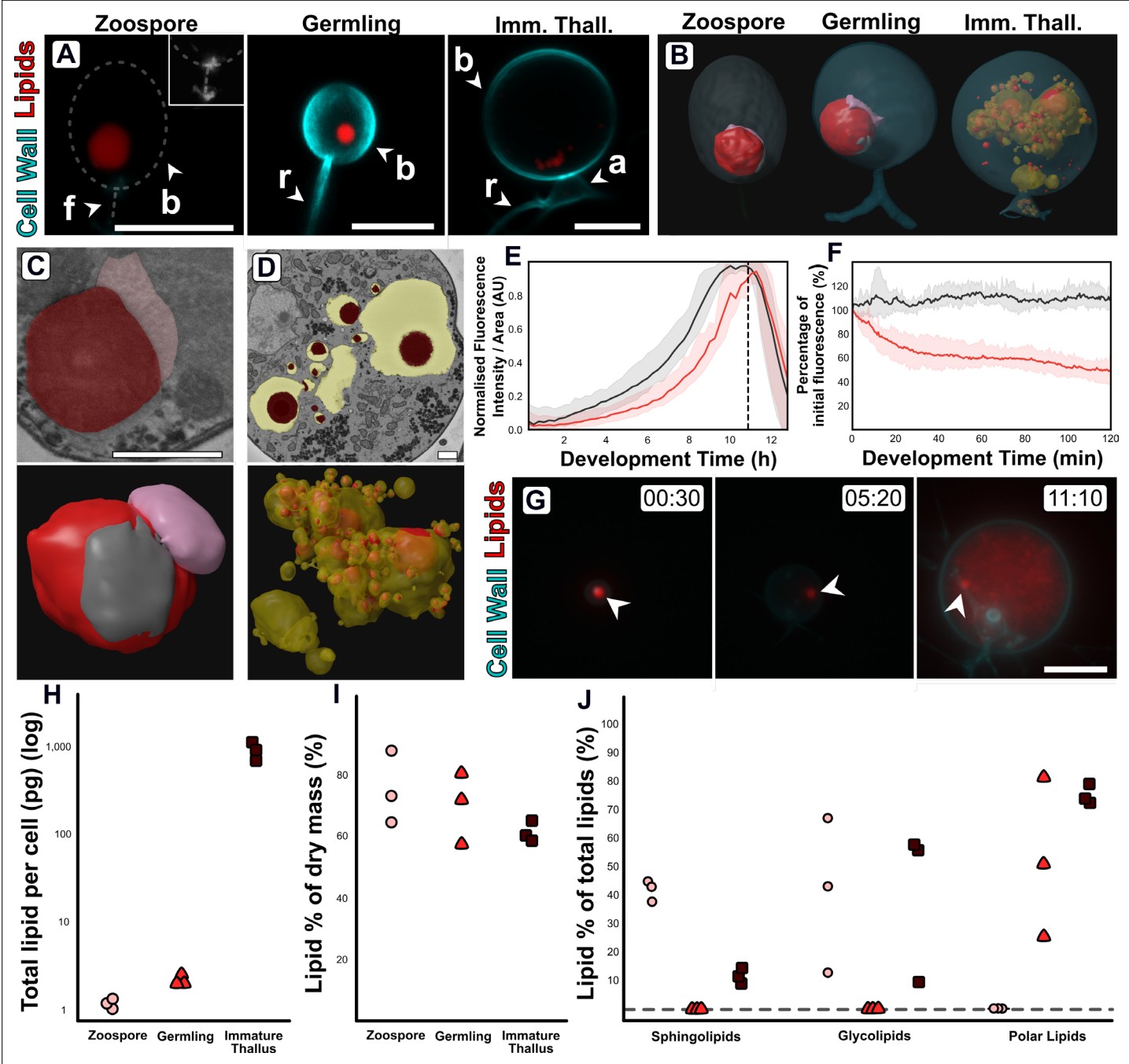

**Figure 4.** Changes in lipid and lipid-associated cell structures occur with transitions between life stages. (**A**) Fluorescent labelling of *Rhizoclosmatium globosum* shows distinct shifts in lipid structures across the chytrid life cycle and cell wall. Dashed line demarks cell boundary where not labelled in the zoospore. Zoospore inset shows precursory cell wall material at the flagellar base contrast-brightness adjusted for visualisation. Apophysis (**a**), cell body (**b**), flagellum (**f**), rhizoid (**r**). Scale bars = 5 µm. (**B**) Representative serial block face scanning electron microscopy (SBF-SEM) reconstructions of lipid globules and lipid-associated structures across chytrid life stages. (**C–D**) Representative single false-coloured SBF-SEM slices (top) and SBF-SEM reconstructions (bottom) of the lipid-rumposome-microbody (LRM) complex from zoospores (also seen in germlings) (**C**) and intravacuolar lipid globules (**D**) from immature thalli. Scale bars = 1 µm. (**E**) Live-cell imaging (n = 5) of *R. globosum* population-level Nile red-stained lipid dynamics. Red = mean lipid fluorescence (±min/max), black = mean total cell area (±min/max), dashed line = mean sporulation time of population. (**F**) Immediately following zoospore settlement, the population-level (n = 5) lipid fluorescence (red) decreases relative to fixed photobleaching control populations (black). (**G**) Live-cell imaging revealed differential lipid dynamics across the chytrid life cycle. Note that the original zoospore lipid globule (arrowhead) remains intact up to the point of lipid anabolism in the immature thallus. Timestamp = HH:MM. Scale bar = 10 µm. (**H–J**) Lipid analysis shows shifts in lipid composition

*Figure 4 continued on next page*

*Figure 4 continued*

of the chytrid life cycle. Lipid quantities as total mass per cell (**H**) and as a percentage of total dry mass (**I**) between chytrid life stages. Changes in lipid fractions were found between chytrid life stages (**J**). Dashed line = below analytical detection.

The online version of this article includes the following source data and figure supplement(s) for figure 4:

**Source data 1.** Data associated with *Figure 4*.

**Figure supplement 1.** Comparison of sporulation times (as a proxy for normal cell development) for dye-labelled chytrid populations (n = 5) imaged by live-cell microscopy, relative to no-dye controls.

of these genes between the life stages overall, however mRNA from five putative chitin synthase genes were abundant during the zoospore stage (*Figure 4A*; *Supplementary file 4* Source Data). Nine genes were only found upregulated in the immature thallus relative to the germling, six of which had >5-fold change increase in abundance. Six genes were not recovered in any the transcriptomes. Interestingly, a putative β-1,6-glucan synthase gene (ORY39038) identified in *Laundon et al., 2020*, as having a possible role in wall formation in *R. globosum* was downregulated in germlings relative to mRNA abundance in the wall-less zoospore stage. Together, this suggests that cell wall formation is a dynamic process throughout the chytrid life cycle, with alternative synthesis enzymes employed at different stages.

## Changes in subcellular lipid-associated structures and lipid composition

Fluorescent labelling and SBF-SEM reconstructions showed that *R. globosum* zoospores and germlings possess the archetypal single lipid globule (Z 0.9 ± 0.6 and G 1.9 ± 1.1 μm³) whereas immature thalli have multiple (68.8 ± 55.2) but smaller (0.5 ± 0.8 μm³) globules scattered throughout the cell body (*Figure 4A–B*; *Video 2*). The lipid globule (red) in the zoospore and germling stages was associated with a posteriorly oriented structure called the rumposome (grey), which is a chytrid-specific organelle putatively associated with cell signalling (*Powell, 1983*), and an anteriorly oriented microbody (pink) that likely functions as a lipid-processing peroxisome (*Powell, 1976*; *Figure 4B–C*; *Video 2*). Together these structures form the lipid-rumposome-microbody (LRM) complex. The rumposome was larger in zoospores than in germlings (Z 0.3% ± 0.0% vs. G 0.1% ± 0.1%, p < 0.001) (*Figure 2*; *Figure 2—figure supplement 5*), indicating increased activity in zoospores. In immature thalli, LRM complexes were not detected. Unlike in zoospores and germlings, the bulk of the lipids in the immature thalli were intravacuolar (89.8% ± 8.5% total lipids) (*Figure 4D*). There was no proportional volumetric difference in lipid fractions determined with SBF-SEM reconstructions between the three life stages (Z 4.3% ± 2.6% vs. G 5.7% ± 3.7% vs. IT 4.0% ± 1.6%, p > 0.05) (*Figure 2B–D*; *Figure 2—figure supplement 5*).

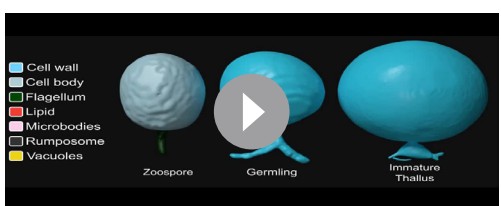

**Video 2.** Structural shifts in lipid globules were observed across *Rhizoclosmatium globosum* life stages, associated with the change from catabolism/conversion to anabolism. Representative serial block face scanning electron microscopy (SBF-SEM) reconstructions of the zoospore, germling, and immature thallus lipid structures for comparison.

https://elifesciences.org/articles/73933/figures#video2

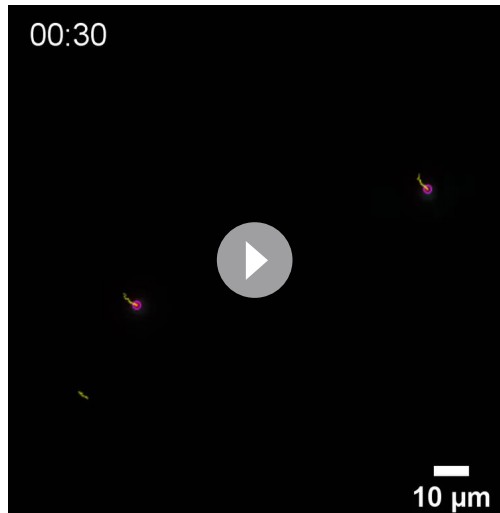

**Video 3.** The zoospore lipid globule remained as an intact structure across the *Rhizoclosmatium globosum* life cycle. Automated particle tracking of lipid globules (red) across the chytrid life cycle. Magenta circles mark individual lipid globules. Yellow track shows particle tracking of the initial lipid globule into the period of lipid anabolism. Cell wall shown in cyan. Timestamp = HH:MM.

https://elifesciences.org/articles/73933/figures#video3

Live-population imaging of Nile Red-labelled storage lipids showed that initially (0–2 hr) the chytrid life cycle was characterised by a decrease (–49.7% ± 9.8%) in lipid fluorescence suggesting that neutral storage lipid catabolism was taking place, before fluorescence increased suggesting that lipid anabolism was occurring up to zoospore release (*Figure 4E–F*; *Figure 4—figure supplement 1*). The initial lipid fluorescence decreased even in the presence of a carbon replete growth medium in line with the non-feeding habit of chytrid zoospores (*Figure 4F*). Live single-cell imaging revealed a similar response as shown at the population level, and additionally showed that the zoospore lipid globule remains intact and detectable until at least the point of visible lipid anabolism in the developing cell when the globule becomes indistinguishable from the new lipids (*Figure 4G*, *Video 3*).

Extraction and quantification of lipids from cells harvested at the major life stages showed shifts in lipid profiles. Individual zoospores possessed 1.2 ± 0.1 pg, germlings 2.2 ± 0.3 pg, and immature thalli 904.5 ± 201.0 pg of lipid per cell (*Figure 4H*), however lipid composition as a percentage of dry mass (Z 74.8% ± 11.8% vs. G 69.5.0% ± 11.5% vs. IT 61.0% ± 3.3%) was similar across the life stages (*Figure 4I*). Sphingolipids were present in both zoospores and immature thalli (Z 41.6% ± 3.6% and IT 11.5% ± 2.6%), but below detection in germlings (*Figure 4J*). Likewise, glycolipids were present in both zoospores (40.7% ± 27.1%) and immature thalli (40.6% ± 27.3%), but below detection in germlings. Conversely, polar lipids were below detection in zoospores yet present in germlings (51.7% ± 27.5%) and immature thalli (74.0% ± 3.4%).

Comparison of the zoospore and germling transcriptomes suggest that the zoospores contain increased abundance of mRNA of genes in KEGG categories associated with peroxisome activity and ABC transporters (*Figure 3C*). Most of the genes identified under the peroxisome category are putatively involved in lipid oxidation and acyl-CoA metabolism (*Figure 3C*; *Figure 3—source data 1*), and therefore likely involved in the catabolic processing of the lipid globule. Lipid reductases were also detected, which have previously been identified with phospholipid anabolism (*Lodhi and Semenkovich, 2014*) and suggest increased phospholipid synthesis for membrane production. ABC transporters are also involved in lipid transport into peroxisomes from lipid stores (*Tarling et al., 2013*). With the quantification of lipids discussed above, these results suggest that glycolipid, and potentially sphingolipid, catabolism from storage lipid utilisation from the globule, possibly via the peroxisome, and polar lipid anabolism are taking place during the transition from zoospore to germling.

We also observed the upregulation of genes associated with fatty acid degradation and peroxisomes in the immature thallus stage compared to the germling stage (*Figure 3F*) coinciding with new lipids being produced (*Figure 4*). The genes were associated with similar acyl-CoA pathways as the zoospore peroxisome category, in addition to alcohol and aldehyde dehydrogenation (*Figure 3C*; *Figure 3—source data 1*). Interestingly, although the peroxisome category was also upregulated in immature thalli, the associated genes were not identical to those in zoospores. Many similar acyl-CoA metabolic signatures were shared (16 genes), but with the addition of alcohol and isocitrate dehydrogenation and superoxide dismutase activity. This suggests that in immature thalli lipid production is driven by an interplay of fatty acid degradation and lipid anabolism, illustrating that some aspects of lipid catabolism and conversion in zoospores are bidirectionally repurposed for anabolism in immature thalli.

## Intracellular trafficking in the apophysis between the rhizoids and cell body

The apophysis is ubiquitous across the Chytridiomycota (*James et al., 2006*), but the function of the structure is poorly understood (*Laundon and Cunliffe, 2021*). Here, we show that the apophysis exhibits high endomembrane density and active intracellular trafficking between the feeding rhizoids and cell body (*Figure 5*). Live-population imaging of FM1–43 labelled endomembrane in *R. globosum* cells (excluding apophysis and rhizoids) showed stability in fluorescence at the beginning of the life cycle (0–2 hr), before a constant increase to the point of zoospore release (*Figure 5A*). Matching this, SBF-SEM reconstruction revealed that immature thalli devoted a larger proportion of cell body volume to endomembrane than zoospores and germlings (Z 2.3% ± 1.5% vs. G 7.6% ± 0.9% vs. IT 13.0% ± 1.8%, p < 0.001), as well as vacuoles (*Figures 2B–D and 5B*; *Figure 2—figure supplement 5*).

The immature thalli SBF-SEM reconstructions showed that apophyses displayed even greater endomembrane than their corresponding cell bodies (apophysis 12.2% ± 5.2% vs. cell body 2.7% ± 0.6%, p < 0.01) (*Figure 5C–D*; *Figure 5—figure supplement 1*; *Video 4*). Apophyses also had

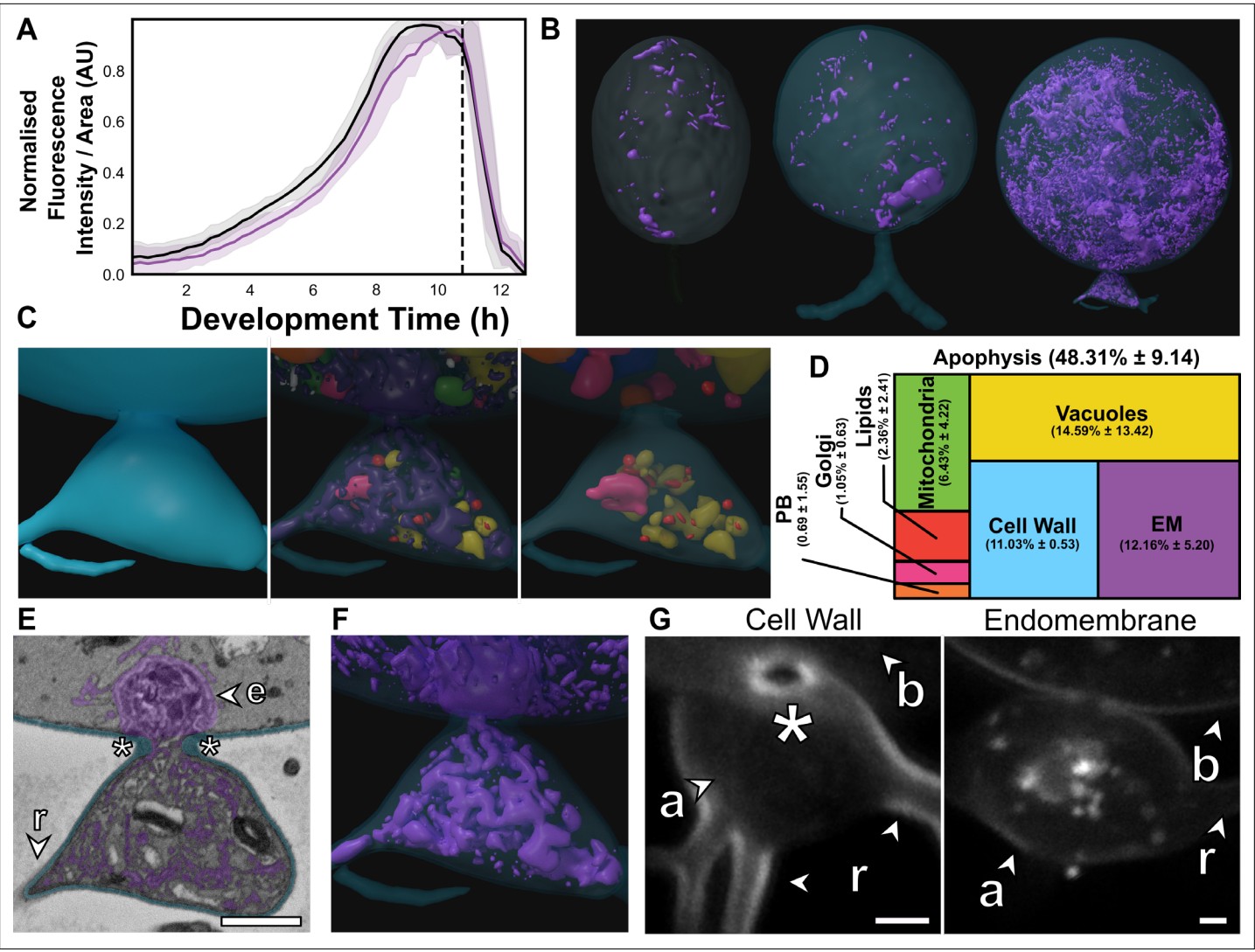

**Figure 5.** The apophysis is a distinct subcellular structure characterised by increased endomembrane trafficking. (**A**) Live-cell imaging (n = 5) of *Rhizoclosmatium globosum* population-level FM1–43-stained endomembrane dynamics. Purple = mean endomembrane fluorescence (±min/max), black = mean total cell area (±min/max), dashed line = mean sporulation time of population. (**B–C**) Representative serial block face scanning electron microscopy (SBF-SEM) reconstructions of endomembrane across chytrid life stages (**B**) and the apophysis from immature thalli (**C**). Volumetric composition of SBF-SEM reconstructions (n = 5) of immature thallus apophyses (**D**). Representative single false-coloured SBF-SEM slice (**E**) and reconstruction (**F**) of the endomembrane and thickened cell wall (asterisk) at the apophysis-cell body junction. Fluorescent labelling of the chitin-rich wall around the apophysis-cell body connecting pore and associated endomembrane structures (**G**). Labels as in *Figure 4A*. All scale bars = 1 µm.

The online version of this article includes the following source data and figure supplement(s) for figure 5:

**Source data 1.** Data associated with *Figure 5*.

**Figure supplement 1.** Comparisons of volumetric proportions of subcellular structures between immature thalli cell bodies and their corresponding apophyses (n = 5).

comparatively more cell wall than the larger cell bodies (A 11.0% ± 0.5% vs. CB 2.4% ± 0.3%, p < 0.01) (*Figure 5D*; *Figure 5—figure supplement 1*). In *R. globosum* the cytoplasm between the apophysis and the cell body is connected via an annular pore (0.40 ± 0.07 µm in diameter) in a distinctive chitin-rich pseudo-septum (*Figure 5E, G*), causing spatial division within the immature thallus cell plan. Live single-cell imaging showed dynamic endomembrane activity in the apophysis linking the intracellular traffic between the rhizoid system and posterior base of the cell body via the pore (*Video 5*). Taken together, we propose that a function of the apophysis is to act as a cellular junction that sorts intra-cellular traffic and channels material from feeding rhizoids through the pseudo-septal pore to the cell

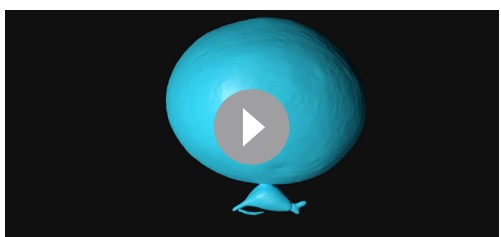

**Video 4.** The *Rhizoclosmatium globosum* apophysis is structurally dominated by endomembrane structures. Representative serial block face scanning electron microscopy (SBF-SEM) reconstruction of a chytrid apophysis from an immature thallus.
https://elifesciences.org/articles/73933/figures#video4

body dedicated for reproduction. Future work could consider examining this structure in more detail, including comparison with other chytrid species across the phylum.

## Developing zoospores in the mature zoosporangium

Understanding zoosporogenesis, including how the thallus differentiates into the next generation of zoospores, is integral to closing the chytrid life cycle. We were unable to achieve a synchronised population of mature zoosporangia, however imaging and sequencing of mixed populations (~4% cells at the mature zoosporangia stage, *Figure 1C*; *Figure 1—figure supplement 1*) still allowed structural characterisation of this life stage, including the SBF-SEM reconstruction of an entire mature zoosporangium containing 82 developing zoospores (*Figure 6*; *Figure 2—figure supplement 2*; *Video 6*). Mature zoosporangia are characterised by internal membrane cleavage (*Figure 6A*) where coenocytic immature cytoplasm and organelles are allocated into nascent zoospores. The volume of the SBF-SEM reconstructed mature zoosporangium was 3651.5 μm$^3$, showing that a single chytrid cell volumetrically increases by more than two orders of magnitude over its entire life cycle. The developing zoospores were flagellate, with the flagellum coiled round the cell body in two neat and complete rotations (*Figure 6B*). Zoospores are held within the cell wall of the zoosporangium during zoosporogenesis (*Figure 6B*), before exiting through the anteriorly oriented discharge pore (an aperture in the cell wall) when developed. During development, the pore is obstructed by a fibrillar discharge plug (49.7 μm$^3$ in volume) (*Figure 6C*).

The single entire zoosporangium reconstruction (*Figure 6B*) allowed the visualisation of developing zoospores in context, but to understand the detailed structural basis of this process it was necessary to reconstruct individual zoospore cells in the zoosporangium at higher resolution for comparison with free-swimming zoospores (*Figure 6D–G*; *Figure 6—figure supplement 1*; *Video 7*). Comparison was also made of transcriptomes from the mature zoosporangia (taken from the mixed populations) with the mRNA content of the free-swimming zoospores (*Figure 6H–J*). Relative to mature free-swimming zoospores, developing zoospores in the zoosporangium displayed an amoeboid morphology and had greater intracellular trafficking, characterised by a larger volumetric proportion of endomembrane (DZ 1.7% ± 0.3% vs. MZ 0.9% ± 0.4%, p < 0.05), vacuoles (DZ 8.4% ± 2.1% vs. MZ 2.3% ± 1.5%, p < 0.001) and the presence of Golgi apparatuses and a vesicle class not observed in mature free-swimming zoospores (*Figure 6D–E*; *Figure 6—figure supplement 1*). Developing zoospores in the zoosporangium also exhibited larger glycogen stores (DV 5.5% ± 1.5% vs. MV 1.6% ± 1.2%, p < 0.01), indicating that glycogen utilisation occurs between the two stages, and a smaller rumposome (DV 0.1% ± 0.0% vs. MZ 1.3% ± 0.0%, p < 0.001) (*Figure 6D–E*) than their mature free-swimming counterparts.

The amoeboid morphology of the developing zoospores was in part a result of possible endocytotic engulfment activity, where vacuoles extended from within the zoospore cell interior to the surrounding interstitial maternal cytoplasm of the zoosporangium (*Figure 6F–G*). Such vacuoles were found in every replicate (2.0 ± 0.7 vacuoles per replicate). The zoospore vacuoles contained electron-dense cargo similar to lipids (*Figure 6F*). The prominence of this proposed engulfment across replicates indicates the possibility that endocytosis is a mode by which resources are trafficked from the maternal cytoplasm into developing zoospores post-cleavage, and that zoospore development does not cease

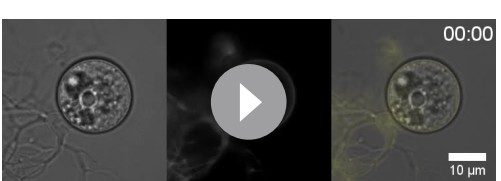

**Video 5.** The *Rhizoclosmatium globosum* apophysis links intracellular trafficking between the rhizoids and cell body. Live-cell imaging of endomembrane dynamics in the chytrid apophysis. The apophysis links endomembrane dynamics between the rhizoid system and thallus. Shown are DIC (left), endomembrane (centre), and overlay (right) channels. Timestamp = MM:SS.
https://elifesciences.org/articles/73933/figures#video5

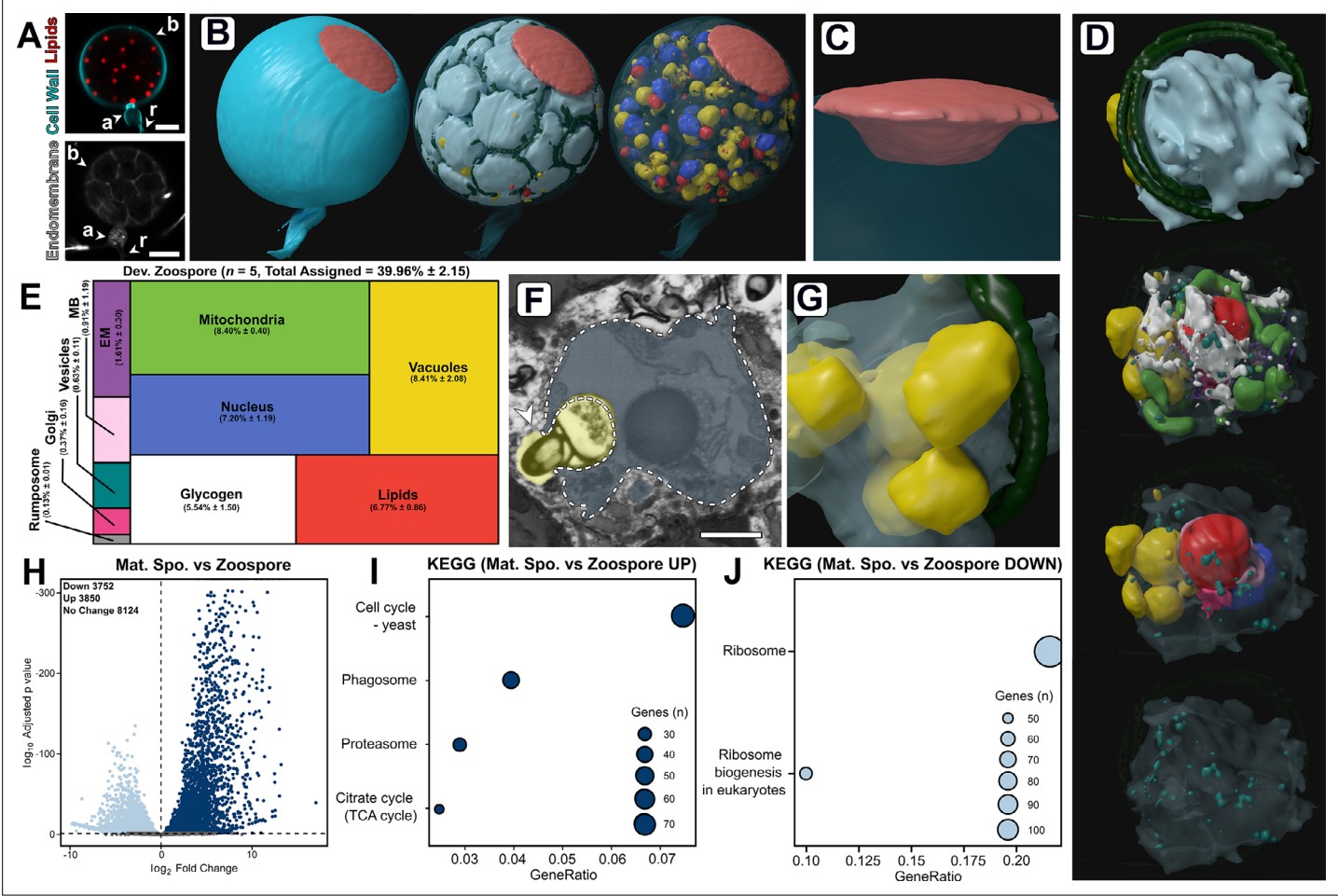

**Figure 6.** Developing zoospores in the zoosporangium displayed an amoeboid morphology with putative endocytotic activity. (**A**) Fluorescent labelling of lipids, cell wall, and endomembrane in a *Rhizoclosmatium globosum* mature zoosporangium. Scale bar = 5 µm. (**B–C**) Serial block face scanning electron microscopy (SBF-SEM) reconstructions of an 82-zoospore containing mature zoosporangium (**B**) highlighting the discharge plug, shown in coral (**C**). (**D**) Representative SBF-SEM reconstructions of a developing zoospore. Organelle colours as in **E**. (**E**) Volumetric composition of SBF-SEM reconstructions of developing zoospores (n = 5). (**F–G**) Representative single false-coloured SBF-SEM slice (**F**) and reconstruction (**G**) of the putative endocytotic vacuoles in developing zoospores. Dashed line delineates the zoospore cell boundary in (**F**). Scale bar = 1 µm. (**H**) Pairwise comparison of differentially expressed genes (DEGs) between mature zoosporangia and the free-swimming zoospore life stage. (**I–J**) Pairwise comparison of significant Kyoto Encyclopedia of Genes and Genomes (KEGG) categories between mature zoosporangia and the free-swimming zoospore life stage.

The online version of this article includes the following source data and figure supplement(s) for figure 6:

**Source data 1.** Data associated with *Figure 6*.

**Figure supplement 1.** Comparisons of volumetric proportions of subcellular structures between developing and mature zoospores (n = 5).

once cleavage has been completed. Notably, developing zoospores did not yet display a detectable ribosomal cluster, as in the free-swimming zoospores (*Figure 2*), indicating that this structure is formed later in zoospore development than captured here. The only KEGG categories higher in free-swimming zoospores than in the mature zoosporangia samples were associated with ribosomes (*Figure 6J*). These different lines of evidence reinforce the importance of maintaining ribosomes in the biology of transcriptionally inactive zoospores and close the chytrid life cycle when considered with our early discussion on the distinctiveness of zoospores in the zoospore-germling transition.

## Discussion

This study into the cellular and molecular biology of *R. globosum* has generated a novel developmental atlas of an archetypal chytrid life cycle, shedding light on the cell patterns of major life stages and

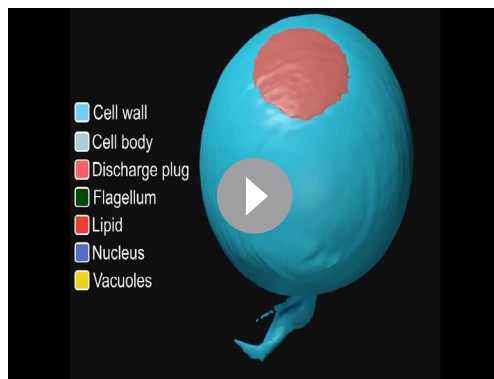

**Video 6.** Serial block face scanning electron microscopy (SBF-SEM) reconstruction of a *Rhizoclosmatium globosum* mature zoosporangium. https://elifesciences.org/articles/73933/figures#video6

the biological processes governing the transitions between them. Our key findings are summarised in *Figure 7*.

In the *R. globosum* zoospore cell body, the ribosome cluster is a distinctive and dominating feature. Historically called the 'nuclear cap', ribosome clusters have been observed in zoospores throughout the Chytridiomycota (e.g. *Koch, 1961*) and in the closely related Blastocladiomycota (e.g. *Lovett, 1963*). *Lovett, 1963*, showed in *Blastocladiella* (Blastocladiomycota), as we show here with *R. globosum*, the ribosome cluster dissipates during the transition between the free-swimming zoospore and germling stages causing the release of the previously contained ribosomes throughout the cell. *Lovett, 1968*; *Lovett, 1963*, related the *Blastocladiella* zoospore ribosome cluster and subsequent dissipation with the biological activity of the cell during the zoospore-germling transition, proposing the role of the cluster was to maintain the ribosomes through the zoospore stage and to spatially isolate the ribosomes to prevent translation occurring until when the cluster dissipates (i.e. the germling stage) and protein synthesis is initiated. Subsequent investigations into transcription and translation in chytrids and the Blastocladiomycota established that these processes are inactive in zoospores which contain maternally provisioned mRNA and dormant ribosomes (*Barstow and Lovett, 1974*; *Jaworski, 1976*; *LéJohn and Lovett, 1966*; *Lovett, 1968*; *Schmoyer and Lovett, 1969*; *Silva et al., 1987*).

Our transcriptome data add molecular detail to understanding the biology of the zoospore-germling transition, with KEGG categories related to ribosome maintenance downregulated and categories associated with translation and biosynthesis upregulated in the germling life stage compared to zoospore mRNA. Similarly, *Rosenblum et al., 2008*, detected high levels of transcripts associated with posttranslational protein modification in *Batrachochytrium dendrobatidis* (*Bd*) zoospores, but low transcriptional activity. Taken together, the chytrid zoospore represents a sophisticated and well-adapted life stage specialised for dispersal to new growth substrates or hosts rather than general metabolism, which is only initiated by the release of the ribosome cluster at the germling stage once favourable conditions are found. The germling stage is characterised by major cell plan remodelling, including rhizoid growth, and concomitant activation of diverse metabolic pathways. Similar upregulation of metabolic pathways has been observed at the transcriptional level associated with conidial germination in dikaryan fungi (*Sharma et al., 2016*; *Zhou et al., 2018*).

Interestingly, we detected the upregulation of proteasome genes in the germling relative to the zoospore mRNA content, which are also necessary for dikaryan germination (*Seong et al., 2008*; *Wang et al., 2011*). A previous study into flagellar retraction in *R. globosum* showed that the internalised flagellum is disassembled and degraded in the germling stage, at least partially by proteasome-dependent proteolysis (*Venard et al., 2020*). Our finding of increased proteasome-associated mRNA may likewise be associated with flagellar degradation and the recycling of redundant zoospore machinery in the germling.

The immature thallus displayed increased cellular and molecular signatures associated with

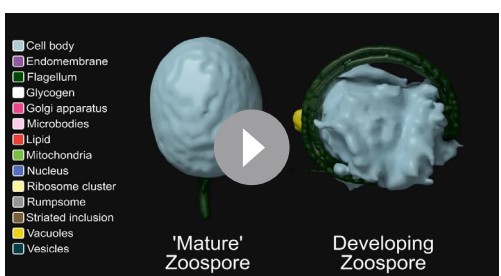

**Video 7.** Developingidual 3D serial block face scanning electron micr zoospores were more amoeboid than mature zoospores in *Rhizoclosmatium globosum*, putatively due to elevated endocytosis and trafficking. Representative serial block face scanning electron microscopy (SBF-SEM) reconstructions of the 'mature' zoospore and developing zoospore life stages for comparison. https://elifesciences.org/articles/73933/figures#video7

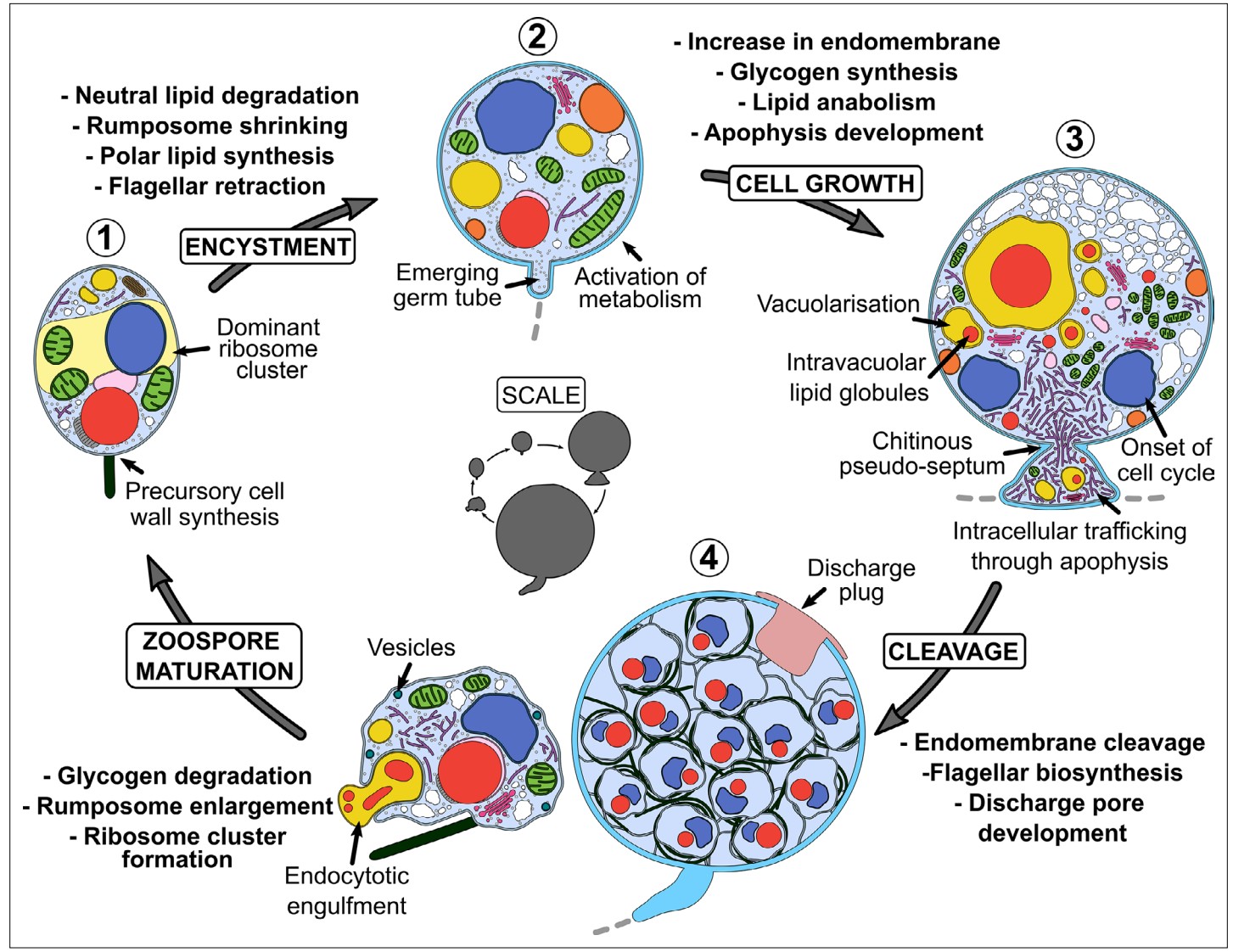

**Figure 7.** Summary of key components of the chytrid cell plan and biological processes associated with the transition between stages in the *Rhizoclosmatium globosum* life cycle. Inner life cycle shows life stages to scale. Grey dashed lines indicate the beginning of the rhizoid system.

the reproductive cell cycle, intracellular trafficking, and protein processing. A key structural development was the vacuolisation of the cell body. Highly vacuolated dikaryan cells (*El Ghaouth et al., 1994*; *Gow and Gooday, 1987*) are associated with diverse cellular processes including general homeostasis, protein sorting, cell cycling, and intracellular trafficking (*Veses et al., 2008*), any of which could at least partially explain the high vacuolisation of immature chytrid thalli. Noticeable in the context of chytrid cell biology however is the increased mRNA for actin-driven cytoskeletal genes, including those assigned to the Arp2/3 complex. The role of actin in vacuolisation and endocytosis has been demonstrated in yeast (*Eitzen et al., 2002*; *Gachet and Hyams, 2005*), similar to our observations here in *R. globosum*. Arp2/3-dependent actin dynamics drive crawling α-motility in some chytrid zoospores when moving freely in the environment (*Fritz-Laylin et al., 2017*; *Medina et al., 2020*) and the presence of animal-like actin components that have been lost in multicellular fungi makes chytrids useful models to investigate the evolution of the fungal cytoskeleton (*Prostak et al., 2021*). Although *R. globosum* zoospores do not crawl, the immature thallus has actin patches, cables, and perinuclear shells (*Prostak et al., 2021*). Here, we show that, for a non-crawling chytrid, actin-associated gene transcripts are increased in immature thalli and that this life stage is associated with high vacuolisation and possible endocytosis, which together could be associated with the early onset of zoosporogenesis.

We did not find any differences in molecular signatures related to cell wall synthesis between the different life stages at the higher categorical level in *R. globosum*, instead we observed individual differentially transcribed genes suggesting that the process is dynamic and complex. Higher levels of putative chitin synthase gene transcripts (e.g. ORY39038) in wall-less zoospores was coupled with the detection of precursory cell wall material at the base of the *R. globosum* flagellum. *Bd* transcriptomes also suggest some chitin synthases to be more abundant in zoospores than in sessile thalli (*Rosenblum et al., 2008*). Chitin synthase activity has been shown to be associated with the *Blastocladiella emersonii* zoospore membrane (*Dalley and Sonneborn, 1982*). Initiation of cell wall synthesis warrants further study and may explain why early chemical inhibition induces phenotypic disruptions to normal development in chytrids (*Laundon et al., 2020*). This emphasises the need to include the wall-less zoospore stage in investigations into chytrid cell wall biology.

This study has highlighted the complexity of lipid dynamics across the *R. globosum* life cycle. Our data show that the volume of the lipid globule, total lipid by volume, and lipid as a percentage of dry mass remain unchanged between zoospores and germlings, yet we observed a shift in lipid type, moving from sphingolipids and neutral glycolipids (likely storage triacylglycerides) to polar lipids (likely membrane-associated phospholipids) between zoospores and germlings. Similarly, during the *B. emersonii* zoospore-germling transition glycolipids decrease and phospholipids increase (*Dalley and Sonneborn, 1982*). Previous research has characterised fatty acid profiles in chytrids (*Akinwole et al., 2014*; *Gerphagnon et al., 2019*; *Rasconi et al., 2020*) and shown differences between chytrid zoospores and sessile thalli of the same species (*Taube et al., 2019*). As the Nile Red emission spectrum undergoes a red shift in increasingly polar environments (*Bertozzini et al., 2011*), we propose that our live-cell data do not quantify the structural degradation of the lipid globule per se but rather biochemical polarisation as neutral storage lipids are catabolised and polar phospholipids are synthesised. The larger volumetric proportion of glycogen stores in developing zoospores over mature free-swimming zoospores also indicates that glycogen catabolism between the two stages contributes to the zoospore energy budget during motility as previously proposed (*Powell, 1979*).

Another novel aspect of our study shows that changes in lipid profiles are apparently coupled with subcellular ultrastructure in *R. globosum*. The enzymatic function of LRM-associated microbodies as lipolytic organelles has been previously proposed (*Powell, 1979*; *Powell, 1977*; *Powell, 1976*), where evidence suggests that enzymatic activity increases following germination (*Powell, 1976*). From our data, this organelle may have bidirectional function and be associated with lipid production (anabolism and conversion) as well as catabolism. A key component of the LRM is the enigmatic chytrid rumposome, which was larger in zoospores than germlings. Previous hypotheses have proposed that this organelle is associated with environmental reception and signal transduction in flagellar regulation (*Dorward and Powell, 1983*). An enlarged rumposome in motile zoospores would support a flagellar role, but its retention in germlings implies additional functions, unless there is a prolonged delay in its degradation. The bulk of lipids in immature thalli during anabolism were intravacuolar and comparable intravacuolar inclusions have been identified in chytrid and dikaryan fungi in the past (*Beakes et al., 1992*; *Bourett and Howard, 1994*; *Lösel, 1990*). Intravacuolar lipid droplets have been previously investigated in yeast but in a catabolic capacity (*van Zutphen et al., 2014*; *Vevea et al., 2015*). Although de novo storage lipid synthesis is associated with the ER (*Vevea et al., 2015*), the vacuoles identified here may cache and aggregate nascent globules as part of the lipid anabolic pathway.

The function of the chytrid apophysis has long been overlooked, despite its ubiquity in the Chytridiomycota (*Powell, 1974*; *Powell and Gillette, 1987*; *Taylor and Fuller, 1980*). Here, we provide evidence that the apophysis acts as a junction for dynamic intracellular trafficking from the multiple branches of the rhizoid network into the central cell body. The localisation of high endomembrane activity in the apophysis and subsequent passage through the annular pore in the pseudo-septum into the cell body implicates this structure as a possible sorting intermediary consolidating the rhizoid network. The ability of multicellular dikaryan fungi to translocate assimilated nutrients through their hyphal network from the site of uptake is sophisticated (*van't Padje et al., 2021*; *Whiteside et al., 2019*), and the observed analogous endomembrane flow from feeding rhizoids to the cell body in chytrids is perhaps not surprising. However, the localisation of high endomembrane activity to the functionally delineated apophysis and through the pseudo-septum into the cell body implicates this structure as a specialised junction.

The pseudo-septation of the apophysis and rhizoids from the cell body is evidence for functional compartmentation (i.e. feeding vs. reproduction) within the thallus of a unicellular fungus. Comparable structures are also present in other chytrid species (e.g. *Barr, 1970*; *Beakes et al., 1992*). Division of multicellular dikaryan fungi by septa, where continuity between distinct cytoplasmic compartments is maintained by septal pores, is integral to multicellularity, cellular differentiation, and resilience (*Bleich-rodt et al., 2015*; *Bleichrodt et al., 2012*). The origin of hyphal septa was a major innovation in fungal evolution (*Berbee et al., 2017*; *Nagy et al., 2020*) occurring at the node shared by hyphal and rhiz-oidal fungi (*Berbee et al., 2017*). The role of the apophysis-cell body pseudo-septum (or an analogous structure) in chytrids in delineating functionally dedicated subcellular compartments may represent an evolutionary precursor to dikaryan septa differentiation and multicellularity. Therefore, investigating the chytrid apophysis is not only important for understanding intracellular trafficking biology in the phylum, but also the evolution of multicellularity more widely across the fungal kingdom.

Our quantitative reconstructions of individual developing zoospores in the zoosporangium and comparison with their free-swimming counterparts have added to understanding the underpinning biology of chytrid zoosporogenesis. Perhaps our most striking finding is the amoeboid morphology of developing zoospores, potentially resulting from endocytotic engulfment, and trafficking structures, suggesting that developing zoospores assimilate material from the maternal cytoplasm post-cleavage. Although dikaryan sporogenesis is complex and diverse, it typically involves the septation of hyphal cytoplasm via cell wall synthesis and dikaryan spores are therefore incapable of such engulfment activity (*Cole, 1986*; *Money, 2016*). Phagocytosis has not been observed generally in the fungi (*Yutin et al., 2009*), however in representatives of the basal Opisthosporidia, wall-less endoparasites such as *Rozella* (Cryptomycota) and *Aphelidium* (Aphelida) have been shown to consume host cytoplasm material via phagocytosis (*Powell et al., 2017*; *Karpov et al., 2014*). Nucleariid amoebae (*Yoshida et al., 2009*), choanoflagellates (*Laundon et al., 2019*), and various animal cell types (*Bayne, 1990*) also exhibit analogous endocytotic engulfment behaviour as we provide evidence for here in chytrid zoospores. Chytrid zoosporogenesis warrants further study, however the apparent conservation may indicate that such engulfment behaviour to assimilate subcellular cargo during sporogenesis of wall-less zoospores existed in the last common ancestor of branching fungi and was lost in dikaryan fungi as spores became walled.

In conclusion, our characterisation of the *R. globosum* life cycle has revealed changes in cell structure and biological processes associated with chytrid development, some of which show analogies in dikaryan fungi and others in 'animal like' cells. As important saprotrophs, parasites, and pathogens, our findings provide information into the cellular processes that underpin the ecological importance of chytrids. In addition, our characterisation of a fungus that retains cellular characteristics from the last common ancestor of branching fungi is a step forward in reconstructing the putative biology of this organism. This study demonstrates the utility of developmental studies with model chytrids such as *R. globosum* and reiterates the need for fundamental biology in investigating the function of chytrid cells.

# Materials and methods

## Culture maintenance

*R. globosum* JEL800 was maintained on peptonised milk, tryptone, and glucose (PmTG) agar plates (*Barr, 1986*) in the dark at 23°C. To collect zoospores, mature plates were flooded with 5 ml PmTG and the cell suspension was passed through a 10 µm cell sieve (43-50010-03, pluriSelect) to remove non-zoospore life stages. Zoospore density was quantified under a Leica DM1000 (10× objective) with a Sedgewick Raft Counter (Pyser SCGI) diluted to 1:1000 and fixed in 0.2% formaldehyde. Zoospores were diluted to a working density of $3 \times 10^6$ ml$^{-1}$ prior to inoculation for all experiments.

## Cell harvesting for SBF-SEM, transcriptomics, and lipid quantification

*R. globosum* was grown to progress through the life cycle and sampled at key time points: 0 hr (zoospore), 1.5 hr (germling), 10 hr (immature thalli), and at 24 hr when the population was a mix of stages including mature zoosporangia (*Figure 1C*, *Figure 1—figure supplement 1*). For zoospores, each replicate was harvested from 10 ml of undiluted cell suspension immediately after plate flooding. For germlings, each culture flask (83.3910, Sarstedt) contained 40 ml of liquid PmTG and

was inoculated with 10 ml of zoospore suspension, incubated for 1.5 hr, and pelleted after scraping the flask with an inoculation loop to dislodge adherent cells. Immature thalli replicates were pooled from 10× culture flasks of 25 ml liquid PmTG inoculated with 50 µl zoospore suspension and incubated for 10 hr. Mixed 24 hr populations containing mature zoosporangia were harvested and strained through a 40 µm cell strainer (11587522, FisherBrand) to remove smaller life stages. All incubations were conducted at 23°C and cells were pelleted at 4692 RCF for 5 min. For SBF-SEM, cell pellets were resuspended and fixed in 2.5% glutaraldehyde in 0.1 M cacodylate buffer pH 7.2. Cell pellets were harvested identically for RNA-Seq (n = 3 biological [biologically independent samples] replicates) with the exception that the supernatant was removed before being flash frozen in liquid nitrogen and stored at –80°C. Sub-samples from cell pellets were taken before freezing and diluted 1:1000, fixed in 0.2% formaldehyde, and stained with FM 1–43 FX (F35355, Invitrogen) to visualise cell membranes in order to qualitatively confirm the synchronicity of cultures under a confocal microscope (see further below) before being processed further (*Figure 1C*; *Figure 1—figure supplement 1*). Cell pellets were harvested for lipid extraction and quantification as per RNA samples (n = 3).

## SBF-SEM imaging and reconstruction

Samples were further fixed in buffered glutaraldehyde, pelleted, and embedded in bovine serum albumin gel. Blocks were processed into resin using a modified protocol by Deerinck and colleagues (https://tinyurl.com/ybdtwedm). Briefly, gel-embedded chytrids were fixed with reduced osmium tetroxide, thiocarbohydrazide, and osmium tetroxide again, before being stained with uranyl acetate and lead aspartate. Stained blocks were dehydrated in an ethanol series, embedded in Durcupan resin, and polymerised at 60°C for 24–48 hr. Blocks were preliminarily sectioned to ascertain regions of interest (ROIs) using transmission electron microscopy (FEI Tecnai T12 TEM). ROIs were removed from the resin blocks and remounted on aluminium pins, which were aligned using scanning electron microscopy (Zeiss GeminiSEM) on a Gatan 3View serial block face microtome and imaged.

Stacks of chytrid cells were acquired at 75 nm z-intervals with an XY pixel resolution of 2 nm (zoospore, germling, and developing zoospore inside a mature zoosporangium), 4 nm (immature thallus), and 8 nm (mature zoosporangium). Although XY pixel size differed between life stages, 2–4 nm resolutions were above the minimum sampling limits for quantitative comparison of reconstructed organelles. Due to the lack of replication (i.e. only one cell was assessed, n = 1), the mature zoosporangium life stage was only considered qualitatively. Acquired stacks were cropped into individual cells (n = 5 biological replicates) and imported into Microscopy Image Browser (MIB) (*Belevich et al., 2016*) for reconstruction. Prior to segmentation, images were converted to 8-bit, aligned, contrast normalised across z-intervals using default parameters, and then were processed with a Gaussian blur filter (sigma = 0.6). Stacks were segmented using a combination of manual brush annotation and the semi-automated tools available in MIB (*Figure 2A*; *Figure 2—figure supplement 1*). Briefly, flagella, lipids, microbodies, nuclei, ribosomal clusters, rumposomes, peripheral bodies, striated inclusions, vacuoles, and vesicles were segmented manually using interpolation every three to five slices where appropriate; the discharge plug, endomembrane, glycogen granules, Golgi apparatuses, and mitochondria were masked by coarse manual brushing and then refined by black-white thresholding; and cell boundaries were segmented using the magic wand tool. All models were refined by erosion/dilation operations and manually curated. Models were also refined by statistical thresholding at size cut-offs for each structure consistent across all life stages (either 500 or 1000 voxels).

Structures were volumetrically quantified within MIB. For visualisation of reconstructed cells .am model files were resampled by 33% in XY and imported as arealists into the Fiji (*Schindelin et al., 2012*) plugin TrakEM2 (*Cardona et al., 2012*), smoothed consistently across life stages, and exported as 3D. obj meshes for final rendering in Blender v2.79. All quantification was conducted on unsmoothed models scaled by 50%. Flagella and rhizoids were excluded from quantification as they are not a component of the cell body, and their total length were not imaged in this study. The unassigned cytosol fraction was defined as the total volume of assigned organelles subtracted from the total cell volume and is inclusive of small structures such as ribosomes, vesicles, and small endomembrane and glycogen objects that could not be confidently assigned and were conservatively excluded. Only endomembrane not considered to be predominantly structural (i.e. an organelle or cell-compartment boundary) was reconstructed in the endomembrane category.

## RNA extraction

RNA was extracted from the cell pellets using the RNeasy extraction kit (74004, Qiagen) following the manufacturer's instructions with minor modifications. Cell pellets were thawed in 600 ml RLT lysis buffer containing 10 μl/ml of 2-mercaptoethanol and lysed at room temperature for 5 min with periodic vortexing. Cell debris was removed by centrifuging at 8000 RCF for 1 min, before the lysate was recovered and passed through a QIA shredder (79656, Qiagen). An equal volume of 100% ethanol was added to the homogenised lysate before being transferred to a RNeasy extraction column. RNA was then extracted following the manufacturer's protocol and included an on-column DNase digestion step using the RNase-Free DNase (79254, Qiagen). RNA was quantified using both a NanoDrop 1000 spectrophotometer (Thermo) and the RNA BR assay kit (Q10210, Invitrogen) on the Qubit four fluorometer (Invitrogen). RNA quality was assessed using the RNA 6000 Nano kit total RNA assay (5067–1511, Agilent) run on the 2100 Bioanalyzer instrument (Agilent).

## Sequencing and bioinformatics

Sequencing was carried out using Illumina NovaSeq 6000 technology and base calling by CASAVA, yielding 20,122,633–23,677,987 raw reads by Novogene (https://www.novogene.com). Raw reads were filtered for adaptor contamination and low-quality reads (ambiguous nucleotides > 10% of the read, base quality < 5 for more than 50% of the read) resulting in 19,665,560–22,917,489 clean reads. Reads were mapped against the JEL800 genome using HISAT2 before DEGs between life stages were determined using DESeq2 as part of the Novogene pipeline (*Love et al., 2014*). Transcriptomic profiles were highly conserved between replicates within each of the three life stages (*Figure 3*; *Figure 3—figure supplement 1*). All further analyses were performed in house in R v3.6.1 (R Core Team) using output from the Novogene analysis pipeline. Shared genes between life history stages were displayed using UpSetR (*Conway et al., 2017*). Volcano plots of DEGs were produced using ggplot2 based upon a conservative threshold of log2FoldChange > 0, padj <0.05. Gene Ontology (GO) and KEGG enrichment analysis was carried out using the enricher function in the R package clusterProfiler v3.12 (*Yu et al., 2012*) with a threshold of padj <0.05. Differentially expressed KEGG categories and GO categories (*Figure 3*; *Figure 3—figure supplements 2–5*) were plotted using the dotplot function. For the purposes of this study, analysis and discussion of KEGG pathways was favoured over GO categories as KEGG pathways allow for a more process-oriented interpretation of activity between the life stages.

## Confocal microscopy of subcellular structures

Cell structures were labelled in a 24 hr mixed population with 5 μM calcofluor white (chitin), 1 μM Nile red (neutral lipid), and 5 μM FM1-43 (membranes). Cells were imaged under a 63× oil immersion objective lens with a Leica SP8 confocal microscope (Leica, Germany). Image acquisition settings were as follows: for cell wall excitation at 405 nm and emission at 410–500 nm (intensity 0.1%, gain 20); for lipids excitation at 514 nm and emission at 550–710 nm (intensity 0.1%, gain 50); and for membranes excitation at 470 and 500–650 nm (intensity 5%, gain 50). All life stages were imaged under identical acquisition settings. Cell wall and lipid images are maximum intensity projections at 0.3 μm z-intervals and membrane images are single optical sections. Confocal images are considered qualitatively.

## Live-cell widefield microscopy

Time-lapse imaging of the development of fluorescently labelled subcellular structures was optimised for LED intensity and dye loads that did not interfere with normal cellular development relative to a no-dye control (*Figure 4E–G*; *Figure 5A*; *Figure 4—figure supplement 1*). Population-level development (n = 5 population [biological] replicates) was imaged using an epifluorescent Leica Dmi8 microscope (Leica, Germany) with a 20× objective lens, and single-cell development with a 63× oil immersion lens. Cell structures were labelled as above, with the exception of 1 μM FM1-43 (membrane). Image acquisition settings were as follows: for cell wall excitation at 395 nm and emission at 435–485 nm (intensity 10%, FIM 55%, exposure 350 ms); for lipids excitation at 575 nm and emission at 575–615 nm (intensity 10%, FIM 55%, exposure 1 s); for membranes excitation at 470 and 500–550 nm (intensity 10%, FIM 55%, exposure 2 s); and bright field (intensity 15, exposure 150 ms). Images were captured using a CMOS Camera (Prime 95B, Photometrics); 500 μl of diluted zoospore suspension was applied to a glass bottom dish and cells were allowed to settle in the dark for 15 min.

After this, the supernatant was removed and 3.5 ml of dye-containing PmTG was added to the dish and imaged immediately. To prevent thermal and hypoxic stress during the imaging period, the dish was placed into a P-Set 2000 CT stage (PeCon, Germany) where temperature was controlled at 22°C by an F-25 MC water bath (Julabo, Germany), and the dish was covered by an optically clear film which permits gas exchange. Single images were taken at 15 min intervals for a total of 18 hr for population-level development, and 50 µm z-stacks (2 µm z-intervals) at an interval of 10 min for a total of 14 hr for single-cell development. Lipid degradation in live, settled zoospores was likewise imaged using 100 µl zoospore suspensions labelled with Nile Red in glass bottom dishes. For comparison, labelled zoospores fixed in 0.2% formaldehyde were also imaged to control for photobleaching. Cells were imaged at 30 s intervals for 2 hr. To visualise endomembrane trafficking in the apophysis, 100 µl of 24 hr mixed PmTG cultures stained with 10 µM FM 1–43 were likewise imaged in glass bottom dishes under a 63× oil immersion objective at 30 s intervals for 30 min.

## Image analysis for live-cell microscopy

Developmental time series of fluorescently labelled subcellular structures were analysed with a custom workflow based around scikit-image 0.16.2 (*van der Walt et al., 2014*) run with Python 3.7.3 implemented in Jupyter Notebook 6.0.3. Briefly, cells were segmented using the bright-field channel by Sobel edge detection (*Kanopoulos et al., 1988*) and Otsu thresholding (*Otsu, 1979*). This mask was used to quantify normalised intensity in the fluorescence channel. Every cell in the population was analysed individually, and then the data aggregated at the population level. For lipid tracking during single-cell development, images from the lipid channel were converted to maximum intensity projections and lipid globules were automatically detected using differences of Gaussian (DoG) detection in the Fiji plugin TrakMate (*Tinevez et al., 2017*). Tracking of the initial lipid globule was conducted using a simple LAP tracker. Lipid tracking is only considered qualitatively.

## Lipid extraction and quantification

Lipids (n = 3 pellet [biological] replicates) were extracted using the Bligh and Dyer method (*Bligh and Dyer, 1959*). Lyophilised culture pellets were submersed in a 2:1:0.8 (v/v/v) methanol (MeOH), dichloromethane (DCM), and phosphate buffer (PB) and sonicated for 10 min in an ultrasonic bath before being centrifuged at 3000 rpm for 2 min. The supernatant was collected, and the pellet was re-extracted twice. The combined supernatant was phase separated via addition of DCM and PB (giving an overall ratio of 1:1:0.9 (v/v/v)) and centrifugation at 3000 rpm for 2 min. The lower solvent phases were extracted prior to washing the remaining upper phase twice with DCM. The three lower solvent phases were collected and gently evaporated under oxygen-free nitrogen (OFN) in a water bath held at 25°C (N-EVAP, Organomation, Berlin, MA). The initial lipid extracts were weighed to quantify total lipid biomass before being dissolved in 9:1 (v/v) DCM:MeOH and loaded onto preactivated silica gel (4 hr at 150°C) columns for fractionation. Lipid fractions were separated by polarity via washing the column with one volume of DCM, followed by one volume of acetone and two volumes of MeOH. Each fraction was collected separately, evaporated to dryness under OFN and weighed. Due to the small sample size, lipid data is only considered qualitatively.

## Statistical analysis

All data were tested for normality and homogeneity assumptions using a Shapiro and Levene's test respectively (threshold p > 0.05). If assumptions could be met, then differences between zoospore, germling, and immature thallus volumetric proportions were assessed using ANOVA followed by Tukey's HSD post hoc testing or, if not, then by a Kruskal-Wallis followed by a Dunn's post hoc test. If a structure was entirely absent from a life stage (e.g. no cell wall in the zoospore stage), then the life stage was eliminated from statistical analysis to remove zero values and the remaining two life stages were compared using a *t*-test or Mann-Whitney U test depending on assumptions, and then the removed life stage qualitatively assigned as different. The differences between cell bodies and apophyses in the immature thallus life stage, and between mature zoospores and developing zoospores, were compared using either a paired *t*-test or a Mann-Whitney U test depending on assumptions. All statistical analysis was conducted using the scipy package (*Virtanen et al., 2020*) run with Python 3.7.3 implemented in Jupyter Notebook 6.0.3.

## Acknowledgements

We thank Chris Neal at the Wolfson Bioimaging Facility (University of Bristol) for SBF-SEM optimisation and imaging acquisition. We would also like to thank Joyce Longcore (University of Maine) for providing *R. globosum* JEL800 from her chytrid culture collection (now curated by the Collection of Zoosporic Eufungi at the University of Michigan https://czeum.herb.lsa.umich.edu). DL was supported by an EnvEast Doctoral Training Partnership (DTP) PhD studentship funded from the UKRI Natural Environment Research Council (NERC grant no. NE/L002582/1). NC, KB, ST, and MC were supported by the European Research Council (ERC) (MYCO-CARB project; ERC grant agreement no. 772584).

## Additional information

### Funding

| Funder | Grant reference number | Author |
|---|---|---|
| Natural Environment Research Council | NE/L002582/1 | Davis Laundon |
| H2020 European Research Council | 772584 | Michael Cunliffe<br>Seth Thomas<br>Kimberley Bird<br>Nathan Chrismas |

The funders had no role in study design, data collection and interpretation, or the decision to submit the work for publication.

### Author contributions

Davis Laundon, Conceptualization, Data curation, Formal analysis, Investigation, Methodology, Software, Validation, Visualization, Writing - original draft, Writing – review and editing; Nathan Chrismas, Data curation, Formal analysis, Methodology, Software, Validation, Visualization, Writing – review and editing; Kimberley Bird, Investigation, Methodology, Writing – review and editing; Seth Thomas, Formal analysis, Investigation, Methodology, Visualization, Writing – review and editing; Thomas Mock, Supervision, Validation, Writing – review and editing; Michael Cunliffe, Conceptualization, Funding acquisition, Project administration, Resources, Supervision, Validation, Writing - original draft, Writing – review and editing

### Author ORCIDs

Davis Laundon http://orcid.org/0000-0002-1508-664X
Nathan Chrismas http://orcid.org/0000-0002-2165-3102
Kimberley Bird http://orcid.org/0000-0002-7244-5960
Michael Cunliffe http://orcid.org/0000-0002-6716-3555

### Decision letter and Author response

Decision letter https://doi.org/10.7554/eLife.73933.sa1
Author response https://doi.org/10.7554/eLife.73933.sa2

## Additional files

### Supplementary files

• Supplementary file 1. Mean volumetric quantities of cellular structures recorded across chytrid life stages.

• Supplementary file 2. Mean numerical quantities of cellular structures recorded across chytrid life stages.

• Supplementary file 3. Mean volumetric percentages and statistical comparisons of cellular structures recorded across chytrid life stages.

• Supplementary file 4. Mean volumetric percentages and statistical comparisons of cell bodies and their corresponding apophyses in immature thalli.

• Supplementary file 5. Mean volumetric percentages and statistical comparisons of free-swimming and developing zoospores.

• Transparent reporting form

• Source code 1. Python script used to quantify population-level fluorescence of developing chytrid cells (*Figures 4 and 5*).

• Source code 2. Python script used to quantify single-cell Nile Red fluorescence of settled chytrid zoospores (*Figure 4*).

## Data availability

All 3D objects, raw datasets associated with figures, image analysis scripts, processed SBF-SEM stacks, and model files, are available for download from Figshare at: https://tinyurl.com/yww6h9d9. Raw sequencing reads are deposited in the Sequence Read Archive (SRA) (PRJNA789147).

The following dataset was generated:

| Author(s) | Year | Dataset title | Dataset URL | Database and Identifier |
|---|---|---|---|---|
| Laundon D, Chrismas N, Bird K, Thomas S, Mock T, Cunliffe M | 2021 | Mapping of a cellular and molecular atlas reveals the basis of chytrid development | https://www.ncbi.nlm.nih.gov/bioproject/PRJNA789147/ | NCBI BioProject, PRJNA789147 |

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
