## [Editor Report]

This data rich manuscript provides highly valuable insights into chytrid development and will accelerate further research on non-dikaryan fungal cell biology.

---

## [Decision Letter]

**Decision letter after peer review:**

Thank you for submitting your article "A cellular and molecular atlas reveals the basis of chytrid development" for consideration by *eLife*. Your article has been reviewed by 2 peer reviewers, and the evaluation has been overseen by a Reviewing Editor and Jürgen Kleine-Vehn as the Senior Editor. The reviewers have opted to remain anonymous.

The reviewers have discussed their reviews with one another, and the Reviewing Editor has drafted this to help you prepare a revised submission. The reviewers pinpoint that the manuscript is very rich in data and that it is at the border of being comprehensible for the reader. In addition, they also uncovered several overstatements. While the reviewers do not request new data, they kindly ask you to substantially revise and improve the description and discussion of your work. Please, see the detailed reviewer's comments below.

*Reviewer #1 (Recommendations for the authors):*

Here Laundon et al., report a cellular "atlas" of the model chytrid Rhizoclosmatium globosum. The data presented include beautiful and informative 3D reconstructions of all four key life stages of this species, as well as transcriptional profiling of matched samples and analysis of lipid compositions. The data were collected from multiple biological replicates and represent a clearly important resource for the community. With this work, the goal of the authors is to link structural descriptions of chytrid morphology with molecular understanding: this is something that the field needs. The authors describe results in three areas that are very interesting to the field. Unfortunately, the evidence provided for these findings does not always support their conclusions. Additionally, discussion of the literature is insufficient as previous work provides crucial context for interpreting the data presented.

1. Zoospores in several chytrids have been shown to be transcriptionally and translationally inactive, this means that the distribution of transcripts are maternally allocated. Although the authors do cite two papers on the topic in the discussion, this is a fundamental concept that might not be in the mind of non-specialist readers and the authors need to introduce and discuss from the beginning (see PMID: 4412066, 1259436, 3571161), as it provides key context for their finding that germlings have a wider range of transcriptional activity as this is consistent with Rg also being transcriptionally silent in the zoospore state. Finally, the language used to describe transcripts found in zoospores (the manuscript refers to them "expressing" particular genes) is confusing given this context.

2. The authors correlated structural changes observed with general KEGG pathway profiles obtained from transcriptomics. Unfortunately it is hard to pin down exactly what the authors are trying to say about this data because their observations are not placed with precision in the context of what is already known about chytrid development, and KEGG pathways are too broad to be very informative. Drawing inferences about chytrid biology from broad KEGG categories and link them to structural observation is not possible with more detailed molecular analysis. This comes up multiple times: (i) Correlation between an increase in endomembrane structures in a compartment and enrichment of KEGG categories of protein processing and ER etc is not enough to link these endomembrane systems with ER. This requires more direct evidence. (ii) High dynamic activity and endomembrane density in the apophysis is not evidence enough by itself to support the claim of the "apophysis acting as a cellular junction that regulates intracellular traffic." (iii) Although different lipid composition between zoospore and germling, and differences in KEGG categories of peroxisome activity on the other suggest important lipid metabolic changes, these correlations are is not hard enough evidence for the authors to call this process as a "biological characteristic" of the transition from zoospore to germling.

3. The claim that zoospores inside the sporangium undergo phagocytosis is not sufficiently supported by the data presented. To date there is only one case in which it a fungus undergoes a process akin to phagocytosis (i.e. Rozella), and finding a phagocytic fungus would be a very exciting result. Unfortunately, the authors provide no direct evidence to support this specific claim as (i) there are many ways one could imagine to explain the shapes seen in the EM data (perhaps the zoospores are squeezed around the objects), and classic work on Allomyces and Blastocladiella zoosporogenesis indicates that cleavage vesicles can be orderly or very irregular before they align in continuous plates (sometimes concomitant with formation of ribosome aggregates), and that these cleavage planes are nearly complete, but not complete yet. (ii) The genes discussed are not specific to phagocytosis, but are used for a wide variety of other functions. Moreover, the authors appear to equate endocytosis and phagocytosis, and although there is some overlap in the proteins used for these processes, they are not equivalent.

4. Although the author's findings about the complex endomembrane system in Rg apophysis is interesting, the details of the images provided do not support their interpretation of it being a "distinct subcellular structure". Such claims require detailed imaging of the "pseudo-septum", similar to what has been shown for "plasmodesmata" in Entophlyctis and Blastocladiella.

Each of the insufficiently supported claims (see above) should be revised. Additionally, the authors should help the reader understand which findings are new in the paper, and which are confirming findings found in the literature as some of these topics have been discussed previously in the literature. Relatedly, it would be useful for the authors to compare their results to those from other species, for example translational activity in dikaryotic spore germination is fleeting and superficial. Are there any genes expressed by both systems? What assumptions have to be met to be able to compare a Dykarian spore with a zoospore?

The Materials and methods should include all details necessary for a peer to be able to replicate their experiments. This is not the case in the current version of the manuscript.

Editorially, the authors may want to consider reorganizing the manuscript. There is so much information included in each figure, it makes it difficult to appreciate and understand their content. For example, figure 2 includes the overview of all of 3D reconstruction, analysis of the volumetric analysis of that reconstruction, an overview of the gene expression experiments, comparisons between stages of expression data, and KEGG analysis of the datasets.

Additional points that should be addressed:

Sometimes "life cycle" and "cell cycle" are used interchangeably (see abstract and page 4 line 90), but then cell cycle is used in the most common way (i.e. cell division).

Please provide catalog numbers in methods.

Please provide RCF for centrifugation (RPM is not enough to determine force).

Line 561: can you clarify "Sub samples from cell pellets were diluted 1:1000, fixed in 0.2% formaldehyde, and" Does this mean the samples were partially thawed and refrozen?

Further details needed (or a reference): Line 568: and embedded in either agar or Bovine Serum Albumin (BSA) gel".

Line 584: "Due to the lack of replication, the mature zoosporangium was only considered 585 qualitatively. A" This is unclear.

Last paragraph of intro could serve well as first paragraph of results.

*Reviewer #2 (Recommendations for the authors):*

The authors are developing and investigating a molecular atlas of the cell for the chytrid Rhizoclosmatium globosum. They are linking transcriptome and lipidome information data to cellular biology that can be observed through SBF-SEM. The detailed investigation allowed a development of a cell atlas and interpretation of cell wall and organelle dynamics.

The authors successfully explored several hypotheses about development in chytrids including that zoospores are provisioned with maternal mRNA. They interpret the composition of spores to include both essential machinery and instructions for cellular replication but also have more host- or substrate-interaction products primed for the cell to condition development on external signals.

They also explore whether cell wall genes are expressed in a fashion that links to the cell wall-less spore stage, and seem to indicate there are at least one chitin synthase with high levels indicating preparation for dynamic growth and wall formation.

The RNASeq analysis/GO enrichment pointed to secondary metabolism enrichment in some of the comparisons, but there is little discussion of what types of genes these might be in the manuscript. Are these NRPS and siderophore or other product producing genes that contribute to that enrichment category?

The work will have an important impact in the field of cell biology of chytrids but also broadly to Fungi and perhaps also comparative biology of Opisthokonts. The detailed reconstruction and examination of the cellular structures in this lineage should be informative to how transitions occurred in the development of septa in other flagellated lineages (eg Blastocladiomycota) and in Dikarya. The development from encysting zoospore to thallus is clearly complex and this study gives a high resolution and a dynamic examination of the processes.

This is a beautiful paper. I am really impressed and intrigued by the candidate genes/pathways implicated to connect the developmental series to the transcriptome and lipidoome results. I don't have any alternative takes on the interpretations presented, but agree with what the authors have presented to the best of my knowledge of these processes.

There are a lot of figures to present the data collected in this manuscript. It is a tour de force integrating methods though is a bit overwhelming on first or second read of the manuscript. But I think the primary and supplemental figures do provide necessary information to convey so I am not sure how I would suggest any other compaction of the presented material.

There was more variation in the lipid fraction estimates for the germling and sometimes zoospore replicates. Does this suggest non-synchronization of the cells or just that there is a lot of variation of size and stage within the timepoint?

I don't have so much expertise to provide a lot of criticism of either the microscopy and reconstruction or the interpretation of the various bodies and organelles during development. I feel the work is presented visually in a great way and the videos are terrific and make the results even more accessible.

---

## [Author Response]

The reviewers have discussed their reviews with one another, and the Reviewing Editor has drafted this to help you prepare a revised submission. The reviewers pinpoint that the manuscript is very rich in data and that it is at the border of being comprehensible for the reader. In addition, they also uncovered several overstatements. While the reviewers do not request new data, they kindly ask you to substantially revise and improve the description and discussion of your work. Please, see the detailed reviewer's comments below.Reviewer #1 (Recommendations for the authors):Here Laundon et al., report a cellular "atlas" of the model chytrid Rhizoclosmatium globosum. The data presented include beautiful and informative 3D reconstructions of all four key life stages of this species, as well as transcriptional profiling of matched samples and analysis of lipid compositions. The data were collected from multiple biological replicates and represent a clearly important resource for the community. With this work, the goal of the authors is to link structural descriptions of chytrid morphology with molecular understanding: this is something that the field needs. The authors describe results in three areas that are very interesting to the field. Unfortunately, the evidence provided for these findings does not always support their conclusions. Additionally, discussion of the literature is insufficient as previous work provides crucial context for interpreting the data presented.

We thank Reviewer 1 for their positive comments and suggestions for improvement to the manuscript. In the revised version of the manuscript (as detailed below) we have modified some of our conclusions and provided further discussion of the literature to improve context for interpreting the data presented.

1. Zoospores in several chytrids have been shown to be transcriptionally and translationally inactive, this means that the distribution of transcripts are maternally allocated. Although the authors do cite two papers on the topic in the discussion, this is a fundamental concept that might not be in the mind of non-specialist readers and the authors need to introduce and discuss from the beginning (see PMID: 4412066, 1259436, 3571161), as it provides key context for their finding that germlings have a wider range of transcriptional activity as this is consistent with Rg also being transcriptionally silent in the zoospore state. Finally, the language used to describe transcripts found in zoospores (the manuscript refers to them "expressing" particular genes) is confusing given this context.

As requested, we have added to the introduction on the biology of zoospores related to transcription/translation inactivity and maternally deposited mRNA (L 77-79) and have included the suggested references in the Discussion section (L 444-445). We have checked and where appropriate revised the manuscript in terms of language used to describe transcripts in zoospores (e.g. 157, 161, 217, 229).

2. The authors correlated structural changes observed with general KEGG pathway profiles obtained from transcriptomics. Unfortunately it is hard to pin down exactly what the authors are trying to say about this data because their observations are not placed with precision in the context of what is already known about chytrid development, and KEGG pathways are too broad to be very informative. Drawing inferences about chytrid biology from broad KEGG categories and link them to structural observation is not possible with more detailed molecular analysis. This comes up multiple times: (i) Correlation between an increase in endomembrane structures in a compartment and enrichment of KEGG categories of protein processing and ER etc is not enough to link these endomembrane systems with ER. This requires more direct evidence. (ii) High dynamic activity and endomembrane density in the apophysis is not evidence enough by itself to support the claim of the "apophysis acting as a cellular junction that regulates intracellular traffic." (iii) Although different lipid composition between zoospore and germling, and differences in KEGG categories of peroxisome activity on the other suggest important lipid metabolic changes, these correlations are is not hard enough evidence for the authors to call this process as a "biological characteristic" of the transition from zoospore to germling.

We have revised the manuscript to limit the proposed correlation between transcriptome data and the structural changes. We have also highlighted aspects of our work that requires future study. To complement the KEGG output, we also provide GOs as supplementary information so as not to add to the already data-rich manuscript (Figure 3 —figure supplement 2-5). Please note that were relevant, we have highlighted specific transcripts and not only relied on KEGG categories.

3. The claim that zoospores inside the sporangium undergo phagocytosis is not sufficiently supported by the data presented. To date there is only one case in which it a fungus undergoes a process akin to phagocytosis (i.e. Rozella), and finding a phagocytic fungus would be a very exciting result. Unfortunately, the authors provide no direct evidence to support this specific claim as (i) there are many ways one could imagine to explain the shapes seen in the EM data (perhaps the zoospores are squeezed around the objects), and classic work on Allomyces and Blastocladiella zoosporogenesis indicates that cleavage vesicles can be orderly or very irregular before they align in continuous plates (sometimes concomitant with formation of ribosome aggregates), and that these cleavage planes are nearly complete, but not complete yet. (ii) The genes discussed are not specific to phagocytosis, but are used for a wide variety of other functions. Moreover, the authors appear to equate endocytosis and phagocytosis, and although there is some overlap in the proteins used for these processes, they are not equivalent.

We have revised the manuscript to limit claim about zoospores and emphasised that future work is needed on this topic. We have included Rozella as highlighted by the reviewer (L 592-596).

4. Although the author's findings about the complex endomembrane system in Rg apophysis is interesting, the details of the images provided do not support their interpretation of it being a "distinct subcellular structure". Such claims require detailed imaging of the "pseudo-septum", similar to what has been shown for "plasmodesmata" in Entophlyctis and Blastocladiella.

We agree that the complex endomembrane system in the apophysis is interesting and is one of the novel aspects of our study. In the revised manuscript, we have limited this claim and proposed future work.

Each of the insufficiently supported claims (see above) should be revised.

We have made the revisions outlined in the review. Please see above.

Additionally, the authors should help the reader understand which findings are new in the paper, and which are confirming findings found in the literature as some of these topics have been discussed previously in the literature.

As requested, we have revised the manuscript throughout.

Relatedly, it would be useful for the authors to compare their results to those from other species, for example translational activity in dikaryotic spore germination is fleeting and superficial. Are there any genes expressed by both systems? What assumptions have to be met to be able to compare a Dykarian spore with a zoospore?

Considering the comment from the Editor that our manuscript is already very rich in data, we have removed the section on Dikaryon spore germination to not add additional analysis/data.

The Materials and methods should include all details necessary for a peer to be able to replicate their experiments. This is not the case in the current version of the manuscript.

As requested, we have reviewed the Materials and methods section of the manuscript.

Editorially, the authors may want to consider reorganizing the manuscript. There is so much information included in each figure, it makes it difficult to appreciate and understand their content. For example, figure 2 includes the overview of all of 3D reconstruction, analysis of the volumetric analysis of that reconstruction, an overview of the gene expression experiments, comparisons between stages of expression data, and KEGG analysis of the datasets.

Figure complexity was also raised by Reviewer 2. We have reviewed all the figures in terms of making the manuscript more comprehensible. In the revised manuscript, Figure 2 has been separated to two Figures (now Figure 2 and 3).

Additional points that should be addressed:Sometimes "life cycle" and "cell cycle" are used interchangeably (see abstract and page 4 line 90), but then cell cycle is used in the most common way (i.e. cell division).

This has been corrected in the revised manuscript.

Please provide catalog numbers in methods.

Catalogue numbers have been added to the methods in the revised manuscript.

Please provide RCF for centrifugation (RPM is not enough to determine force).

RCF values have been added to the revised manuscript.

Line 561: can you clarify "Sub samples from cell pellets were diluted 1:1000, fixed in 0.2% formaldehyde, and" Does this mean the samples were partially thawed and refrozen?

No, the sub-samples were taken from the pellets before freezing. This has been clarified in the revised manuscript (L 645).

Further details needed (or a reference): Line 568: and embedded in either agar or Bovine Serum Albumin (BSA) gel".

This has been clarified in the revised manuscript (L 651-652).

Line 584: "Due to the lack of replication, the mature zoosporangium was only considered 585 qualitatively. A" This is unclear.

This sentence has been revised for clarity (L 668).

Last paragraph of intro could serve well as first paragraph of results.

The last paragraph of the intro provides a methods summary and so we think important this comes before the Results section.

Reviewer #2 (Recommendations for the authors):The authors are developing and investigating a molecular atlas of the cell for the chytrid Rhizoclosmatium globosum. They are linking transcriptome and lipidome information data to cellular biology that can be observed through SBF-SEM. The detailed investigation allowed a development of a cell atlas and interpretation of cell wall and organelle dynamics.The authors successfully explored several hypotheses about development in chytrids including that zoospores are provisioned with maternal mRNA. They interpret the composition of spores to include both essential machinery and instructions for cellular replication but also have more host- or substrate-interaction products primed for the cell to condition development on external signals.They also explore whether cell wall genes are expressed in a fashion that links to the cell wall-less spore stage, and seem to indicate there are at least one chitin synthase with high levels indicating preparation for dynamic growth and wall formation.The RNASeq analysis/GO enrichment pointed to secondary metabolism enrichment in some of the comparisons, but there is little discussion of what types of genes these might be in the manuscript. Are these NRPS and siderophore or other product producing genes that contribute to that enrichment category?

Because the manuscript is already considered data rich, we have not added further on secondary metabolism. However, full data are provided as supplementary.

The work will have an important impact in the field of cell biology of chytrids but also broadly to Fungi and perhaps also comparative biology of Opisthokonts. The detailed reconstruction and examination of the cellular structures in this lineage should be informative to how transitions occurred in the development of septa in other flagellated lineages (eg Blastocladiomycota) and in Dikarya. The development from encysting zoospore to thallus is clearly complex and this study gives a high resolution and a dynamic examination of the processes.This is a beautiful paper. I am really impressed and intrigued by the candidate genes/pathways implicated to connect the developmental series to the transcriptome and lipidoome results. I don't have any alternative takes on the interpretations presented, but agree with what the authors have presented to the best of my knowledge of these processes.

We thank Reviewer 2 for their positive comments.

There are a lot of figures to present the data collected in this manuscript. It is a tour de force integrating methods though is a bit overwhelming on first or second read of the manuscript. But I think the primary and supplemental figures do provide necessary information to convey so I am not sure how I would suggest any other compaction of the presented material.

As discussed above with Reviewer 1, in the revised version of the manuscript we have updated the figure. Figure 2 has been separated to two figures (now Figures 2 and 3).

There was more variation in the lipid fraction estimates for the germling and sometimes zoospore replicates. Does this suggest non-synchronization of the cells or just that there is a lot of variation of size and stage within the timepoint?

Based on our microscope assessments of the life stages (Figure 1 —figure supplement 1), we are confident that stages 1 (zoospore), 2 (germling) and 3 (immature thallus) were synchronized.

I don't have so much expertise to provide a lot of criticism of either the microscopy and reconstruction or the interpretation of the various bodies and organelles during development. I feel the work is presented visually in a great way and the videos are terrific and make the results even more accessible.

We thank Reviewer 2 again for their supportive and positive comments.